# Transcriptome 3′end organization by PCF11 links alternative polyadenylation to formation and neuronal differentiation of neuroblastoma

Anton Ogorodnikov[1,2,3,16], Michal Levin[1,2,3], Surendra Tattikota[1,2,3], Sergey Tokalov[1,2,3], Mainul Hoque[4], Denise Scherzinger[5], Federico Marini[3,5], Ansgar Poetsch[6,7,8], Harald Binder[9], Stephan Macher-Göppinger[10], Hans Christian Probst[11,12], Bin Tian[4], Michael Schaefer[13], Karl J. Lackner[2], Frank Westermann [14] & Sven Danckwardt[1,2,3,15]

Diversification at the transcriptome 3′end is an important and evolutionarily conserved layer of gene regulation associated with differentiation and dedifferentiation processes. Here, we identify extensive transcriptome 3′end-alterations in neuroblastoma, a tumour entity with a paucity of recurrent somatic mutations and an unusually high frequency of spontaneous regression. Utilising extensive RNAi-screening we reveal the landscape and drivers of transcriptome 3′end-diversification, discovering PCF11 as critical regulator, directing alternative polyadenylation (APA) of hundreds of transcripts including a differentiation RNA-operon. PCF11 shapes inputs converging on WNT-signalling, and governs cell cycle, proliferation, apoptosis and neurodifferentiation. Postnatal PCF11 down-regulation induces a neurodifferentiation program, and low-level PCF11 in neuroblastoma associates with favourable outcome and spontaneous tumour regression. Our findings document a critical role for APA in tumorigenesis and describe a novel mechanism for cell fate reprogramming in neuroblastoma with potentially important clinical implications. We provide an interactive data repository of transcriptome-wide APA covering > 170 RNAis, and an APA-network map with regulatory hubs.

[1] Posttranscriptional Gene Regulation, Cancer Research and Experimental Haemostasis, University Medical Centre Mainz, Mainz 55131, Germany. [2] Institute for Clinical Chemistry and Laboratory Medicine, University Medical Centre Mainz, Mainz 55131, Germany. [3] Centre for Thrombosis and Haemostasis (CTH), University Medical Centre Mainz, Mainz 55131, Germany. [4] Rutgers New Jersey Medical School, Newark, NJ 07103, USA. [5] Institute of Medical Biostatistics, Epidemiology and Informatics, University Medical Centre Mainz, Mainz 55131, Germany. [6] Max-Planck-Institute for Heart and Lung Research, Bad Nauheim 61231, Germany. [7] Institute for Plant Biochemistry, Ruhr-University Bochum, Bochum 44801, Germany. [8] School of Biomedical & Healthcare Sciences, Plymouth University, Plymouth PL4 8AA, United Kingdom. [9] Institute of Medical Biometry and Statistics, Faculty of Medicine and Medical Center—University of Freiburg, Freiburg 79104, Germany. [10] Institute for Pathology, University Medical Centre Mainz, Mainz 55131, Germany. [11] Institute for Immunology, University Medical Centre Mainz, Mainz 55131, Germany. [12] Research Center for Immunotherapy (FZI), University Medical Centre Mainz, Mainz 55131, Germany. [13] Department of Anaesthesiology and Research Centre Translational Neurosciences, University Medical Centre Mainz, Mainz 55131, Germany. [14] Division of Neuroblastoma Genomics, German Cancer Research Centre (DKFZ), Heidelberg 69120, Germany. [15] German Centre for Cardiovascular Research (DZHK), Mainz 55131, Germany. [16] Present address: McManus Laboratory, University of California San Francisco (UCSF), San Francisco, CA 94143, USA. Correspondence and requests for materials should be addressed to S.D. (email: Sven.Danckwardt@unimedizin-mainz.de)

Neuroblastomas are the most common solid tumour in infants accounting for ~15% of all cancer deaths in children. They arise from incompletely committed precursor cells derived from neural crest tissues, and can present as tumour lesions in the neck, chest, abdomen or pelvis. The clinical presentation is heterogeneous, ranging from an asymptomatic tumour disease to a critical illness as a result of local invasion, or as widely disseminated disease. Remarkably, this tumour entity is generally characterised by a lack of recurrent somatic mutations, and exhibits one of the highest proportions of spontaneous and complete regression of all human cancers by as yet unknown mechanisms[1,2].

Next-generation RNA sequencing has led to the discovery of a perplexingly complex metazoan transcriptome architecture arising from the alternative use of transcription start sites, exons and introns and polyadenylation sites[3,4]. The combinatorial use, and incorporation, of such elements into mature transcript isoforms considerably expands genomic information and is subject to dynamic spatial and temporal modulation during development and adaptation. Recently, diversification of the transcriptome at the 3′ end through alternative polyadenylation (APA) evolved as an important and evolutionarily conserved layer of gene regulation[5]. APA results in transcript isoforms varying at the RNA 3′ end, which can encode proteins with profoundly distinct functions or regulatory properties. Furthermore, APA can affect the mRNA fate via the inclusion or exclusion of regulatory elements[6].

Constitutive RNA 3′end maturation relies on a complex macromolecular machinery, which catalyses endonucleolytic cleavage and polyadenylation of pre-mRNA molecules[7]. This involves the assembly of four multicomponent protein complexes (CPSF, CSTF, CFIm and CFIIm)[8] on the pre-mRNA at dedicated processing sites[9]. Differential expression of individual complex components can direct APA resulting in transcript isoforms with alternative 3′ ends[10,11]. In addition, other mechanisms including epigenetic events can affect APA, illustrating a previously unanticipated complex crosstalk between various cellular processes in the control of transcriptome diversity[12]. Interestingly, BARD1, one of the few factors affected by recurrent somatic mutations in neuroblastoma[13] (beyond MYCN, ALK and PHOX2B)[2], forms a heterodimer with CSTF subcomponents to modulate mRNA 3′ end processing[14]. Although difficult to detect by standard high-throughput profiling techniques[15], dynamic changes at the transcriptome 3′ end are prevalent[16,17] (Fig. 1a). They are often associated with differentiation and dedifferentiation processes[11,18]. However, the underlying mechanisms and functional consequences for development and disease remain poorly understood[19].

Here we identify extensive transcriptome 3′end architecture alterations during neuroblastoma differentiation. By combining a genome-wide high-throughput analysis with a comprehensive RNAi screening targeting more than 170 potential components involved in the definition of RNA 3′ ends, we delineate the dynamic landscape of and explore mechanisms influencing transcriptome 3′end diversification in this tumour entity. We identify PCF11, a CFIm complex component involved in transcription termination and RNA 3′end maturation[20,21], as a critical regulator pervasively directing APA of hundreds of transcripts in neuroblastoma. By generating and applying an inducible short hairpin RNA (shRNA) mouse model targeting PCF11 complemented by studies of neuroblastoma patient samples, we discover an unexpected critical role for PCF11-dependent APA regulation in neuronal differentiation with potentially important implications for spontaneous tumour regression.

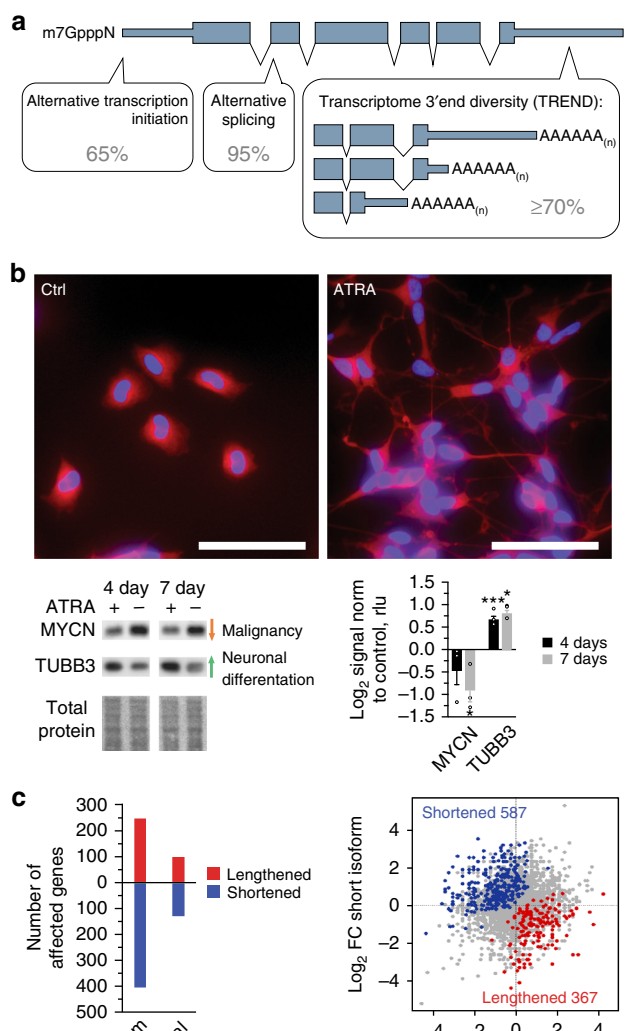

**Fig. 1** Pervasive alteration of transcriptome 3′end diversification (TREND) in childhood neuroblastoma. **a** The genome complexity is considerably expanded by co- and post-transcriptional processes. Various mechanisms including alternative polyadenylation (APA) can result in transcriptome 3′ end diversification (TREND) affecting the coding sequence and/or 3′ untranslated regions (UTR). More than 70% of all genes expressed have alternative 3′ ends, thus shaping the transcriptome complexity to a similar extent as alternative splicing (95%) or alternative transcription initiation (65%). **b** BE(2)-C neuroblastoma model for all-*trans* retinoic acid (ATRA) induced neurodifferentiation (scale bar 100 μm). Western blot of molecular markers reflecting neuronal differentiation in response to ATRA treatment (mean ± s.e.m. for three replicates, one-sided *t*-test *p*-value, *p < 0.05, ***p < 0.001; cells are stained with an antibody directed against TUBB3 (red), nuclei are stained with DAPI (blue)). **c** Differentiation results in a widespread TREND regulation leading to the expression of lengthened (red) and shortened (blue) transcript isoforms, mainly affecting the 3′UTR ('tandem'; also depicted in the scatter plot on the right) compared to relatively fewer events affecting the open reading frame ('internal'; FC fold change)

## Results

**Massively deregulated TREND in neuroblastoma.** To explore the dynamics of transcriptome 3′end diversity (TREND) in the context of neuroblastoma biology (Fig. 1a) we used a cellular model system (BE(2)-C), which faithfully recapitulates critical

features of this type of malignancy in children. A subset of these tumours can be differentiated into benign lesions by employing retinoid drugs (such as all-*trans* retinoic acid (ATRA)), which is standard of care in children with high-risk neuroblastoma[22]. Exposing BE(2)-C cells to ATRA results in neuronal differentiation phenotypically and molecularly (Fig. 1b). By employing a tailored approach based on RNA 3′ region extraction and deep sequencing[15], we observed a significant change of transcripts with alternative 3′ ends upon BE(2)-C differentiation with a trend towards shorter transcript isoforms, predominantly affecting 3′ untranslated regions (UTRs) (Fig. 1c).

Thus, in neuroblastoma, undifferentiated neuronal cells do not only display typical properties of malignant tumour cells (i.e. absence of neurites and MYCN overexpression, Fig. 1b); they also express a substantial fraction of shortened and lengthened transcript isoforms compared to a differentiated state. This invites speculations that deregulated TREND may be functionally linked to the dynamically regulated programme shown here (Supplementary Table 1).

**PCF11 drives TREND by directing APA in neuroblastoma.** We next sought to identify drivers of TREND in neuroblastoma by applying RNAi screening coupled to a newly designed high-throughput sequencing approach suited to capture polyadenylated transcript 3′ends of numerous experimental conditions in a highly multiplexed fashion (TRENDseq; Fig. 2a). We depleted 174 proteins including all known factors involved in pre-mRNA 3′end cleavage and polyadenylation in eukaryotes[8] and selected key factors regulating transcriptional activities, splicing, RNA turnover and other functions[23–32], which could directly or indirectly modulate TREND (Supplementary Table 2)[5]. Probing the efficiency of RNAi-mediated depletion revealed a dropout rate (i.e. poor RNAi effect) below 5% (Supplementary Figure 1b). Applying TRENDseq, we identified in our screening in total 9168 TREND-events (out of 20,156 expressed transcripts, Fig. 2b) corresponding to more than 3600 genes significantly affected by TREND in neuroblastoma cells (BH-adjusted *p*-value < 0.05, Fig. 2c). Upon depletion, almost all tested factors show TREND regulation (Fig. 2b, c) with key factors affecting up to 1400 genes (e.g. NUDT21, CPSF6 and PCF11, Supplementary Table 2; average effect 130 genes). Directly comparing the effects quantitatively and qualitatively, we observe that (i) TREND is predominantly driven by components of the RNA 3′end processing machinery (Fig. 2b, c) and that (ii) their function is mainly non-redundant (Fig. 2c, Supplementary Figure 2a). However, a significant proportion of TREND is controlled by components involved in transcription and other co- and post-transcriptional events (e.g. splicing and RNA turnover) or epigenetic modification—to a quantitatively comparable extent as individual splicing factors affect alternative splicing[33]. Interestingly, our screening also identifies TREND regulation to be caused by factors involved in genome surveillance or known to drive tumour suppressive programmes (e.g. TP53), and other processes involved in the coupling between oncogenic signals and 3′end processing (such as BARD1[14]; details on TREND-affected targets, regulated Gene Ontology (GO) terms and executing TREND regulators are provided online, TREND-DB: http://shiny.imbei.uni-mainz.de:3838/trend-db).

Applying a phylostratigraphy approach we find that almost all transcripts showing a dynamic regulation at the 3′ end are encoded by ancient genes (gene age > 450 million years), as most of the executing TREND regulators (Supplementary Figure 2b,c). Phylogenetically conserved genes control basic processes and are more likely associated with overt phenotypes when deregulated[34]. Interestingly, depletion of numerous TREND regulators results in TREND of targets enriched with cancer-associated genes. Most of

these TREND regulators belong to the RNA cleavage and polyadenylation machinery ($p = 0.0195$), while other functional categories (as a whole) are not significantly enriched (Fig. 2c (inlet), Supplementary Table 3). This suggests that cleavage and polyadenylation factors control TREND in a conserved manner with several of them potentially playing important roles in cancer. Among the top three drivers of TREND, which regulate several hundreds of genes (Supplementary Table 2), we identify components of the CFIm and CFIIm complexes (CPSF6, NUDT21 and PCF11) belonging to the RNA 3′end cleavage and polyadenylation machinery[35] (Fig. 2b, c). Interestingly, unlike the depletion of many other factors screened, their depletion directs TREND in an almost exclusive unidirectional manner resulting in uniformly shortened (CPSF6 and NUDT21) or lengthened (PCF11) transcript isoforms compared to control knockdowns (Supplementary Figure 2d). Thus, in (undifferentiated) neuroblastoma PCF11 promotes proximal polyadenylation site choice, while CPSF6 and NUDT21 facilitate processing at distal polyadenylation sites.

To further explore the functional hierarchy of TREND-regulators we made use of the high reproducibility of TRENDseq (Supplementary Figure 3a). This allowed us to construct a high-confidence network of TREND-regulators ('APA-network map') to visualise their synergistic and antagonistic actions (considering the genes affected and directionality of transcript isoform regulation i.e. shortening or lengthening; Supplementary Figure 3b). Remarkably, the clustering observed in this analysis corresponds to known protein complexes involved in RNA processing[8]. Further it uncovers a strong antagonistic effect between NUDT21 and CPSF6 (CFIm components) and PCF11 (CFIIm complex component). This suggests that they play an important regulatory role in the global organisation of the transcriptome 3′ end in neuroblastoma. Aberrant expression of these components can thus lead to profound TREND-perturbations by affecting APA most quantitatively.

To further define the role of TREND-regulators in the clinically relevant context studied here, we examined their regulation in neuroblastoma upon ATRA differentiation. We probed the protein abundances of all 3′end processing factors and further key candidates by western blotting. Among 82 proteins profiled (Supplementary Figure 4) we observe PCF11 to be most strongly affected (that is a 1.9-fold downregulation upon differentiation, Fig. 3a) compared to other potent TREND-regulators (e.g. NUDT21 and CPSF6). Notably, co-depletion of PCF11 together with other key TREND-regulators establishes PCF11 as a dominant factor in the functional hierarchy of transcriptome 3′ end diversification (Supplementary Figure 5a). PCF11 acts as a dominant repressor of APA at distal sites (i.e. promotes processing at proximal sites), and counteracts the repression of APA at proximal sites executed by CPSF6. Accordingly, PCF11 depletion leads to massive lengthening of the transcriptome, predominantly by de novo activation of previously un-annotated polyadenylation sites far downstream in the 3′ flanking sequence (Supplementary Figure 3a,c). PCF11-directed APA thereby results in a significant production of transcript isoforms harbouring 'new' regulatory elements in their 3′ UTRs (such as miRNA- or RBP-binding sites). This in turn can influence important cellular functions e.g. by modulating the protein output of the affected target RNAs[36–38]. Altogether, PCF11 thus represents a global key driver of TREND by directing APA in the BE(2)-C neuroblastoma model system in a direct and most efficient manner (Fig. 3a, Supplementary Figure 5a-c); PCF11 downregulation reshapes the transcriptome 3′end architecture reminiscent to that observed in mature neurons that typically show a more prevalent expression of long transcript isoforms[39] (Fig. 2b, c, Supplementary Figure 3c). However, we also note that PCF11 is likely not the only factor

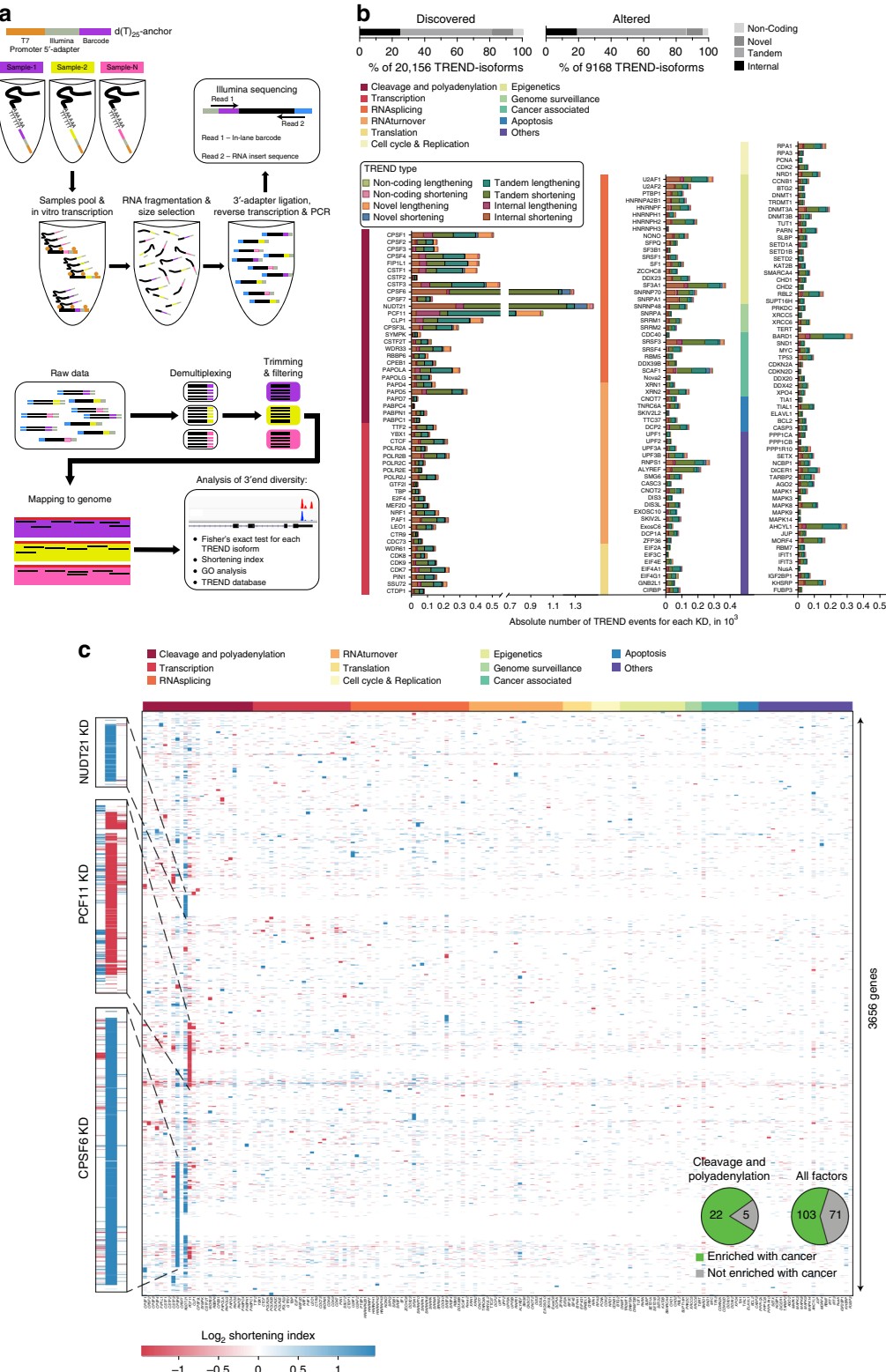

regulated during neurodifferentiation (Fig. 3a). This corresponds to the mixed TREND-pattern of transcript lengthening and shortening during neurodifferentiation in the simplistic neuroblastoma BE(2)-C model (Fig. 1c).

**PCF11-directed APA controls WNT and neurodifferentiation.** The pervasive mode of PCF11 in directing TREND suggests an

important role in basic cellular programmes, including a potential function in tumorigenesis (Supplementary Table 2, cancer enrichment score $2.98 \times 10^{-9}$, Supplementary Figure 6a). To further substantiate this in the context of neuroblastomas, we profiled the proteome of BE(2)-C cells upon PCF11-depletion using mass spectrometry. We identified 5903 proteins out of which 330 showed a regulation of more than 1.5-fold ($p$-value < 0.05). Merging this with the data set of APA-regulated transcripts

**Fig. 2** Massive RNAi screening reveals key regulators of TREND in neuroblastoma with a potential role in tumorigenesis. **a** TRENDseq library preparation (top) and bioinformatical pipeline (bottom) for highly multiplexed genome-wide analysis of the TREND-landscape. The Integrated Genomics Viewer (IGV) of two sequence read alignments (lower right corner) illustrates two distinct TREND-phenotypes that correspond to a more prevalent distal (condition shown in red; reflecting 'lengthening' compared to the respective other condition) or proximal polyadenylation (condition shown in blue; reflecting 'shortening' compared to the respective other condition), respectively. **b** Relative proportion of TREND types discovered in neuroblastoma BE(2)-C (upper left diagram), and isoforms effectively altered upon RNAi depletion of at least 1 out of 174 putative TREND regulators (upper right diagram). 'Tandem' and 'Internal' events affect annotated 3' UTRs or protein C termini, 'Novel' assign transcript isoforms exceeding the annotated gene 3' end and 'Non-coding' depict TREND alterations affecting non-coding RNAs. Individual TREND regulation per RNAi candidate is shown in the bar diagram (the colour code indicates the functional category to which the depleted factor belongs to, see also Supplementary Table 2). **c** Landscape of TREND upon RNAi (KD knockdown) of 174 individual putative TREND regulators (x-axis; functional categories to which the RNAi candidates belong to are the same indicated for **b**; see Supplementary Figure 2a). Each spot in the heat map reflects a gene significantly affected by TREND (the colour code in the heat map indicates the directionality of TREND; for example, a negative shortening index indicates a relatively more prevalent lengthened transcript isoform expressed by the respective gene upon depletion of the respective TREND regulator). Inlet: pie chart showing overrepresentation of cancer-associated genes affected by TREND upon depletion of 3'end processing components (for individual p-values see Supplementary Table 3)

---

upon PCF11 depletion, we observed 54 regulated genes (Supplementary Table 4) enriched in GO terms related to 'neuro-differentiation' (Fig. 3b). Thus, PCF11 depletion regulates APA and protein output of a module of transcripts controlling features of the cellular process i.e. neurodifferentiation, in which we observed the downmodulation of PCF11 (Fig. 3a).

We next validated representative APA-affected targets reflecting the discovered signalling pathways and GO terms (Fig. 3b, Supplementary Figure 6a). We focused on targets that (i) are involved in neuronal and brain development (IGF1R[40]), (ii) have previously been associated with signal transduction in neural cells or neuroblastoma regression (AES, GNB1[41]) or (iii) play a role in other pathways more globally involved in tumorigenesis and the endoplasmic reticulum (ER) stress response (EIF2 signalling). We observed that PCF11-directed APA results in a significant upregulation of IGF1R, EIF2S1 and AES protein abundance, while GNB1 is downregulated upon PCF11 depletion (Fig. 4a). These changes are likely functional as we also noted the expected downstream alterations of the IGF1R, PI3K/Akt and ER stress response signalling pathways. In addition, some of the observed changes upon PCF11 depletion (i.e. activation of the PI3K/Akt pathway) are surprisingly consistent with their reported role for neuroblastoma differentiation[42]. Most notably however, all four validated APA-affected targets (IGF1R, EIF2S1, AES and GNB1) constitute a highly enriched protein interaction network impinging on WNT signalling (i.e. beta-catenin (CTNNB1), p-value $1.86 \times 10^{-4}$, Fig. 4b). This is consistent with the predicted function of the entire set of APA-affected target genes regulated by PCF11 depletion, which show a role in WNT signalling (Supplementary Figure 6a).

**PCF11-directed APA drives neuroblastoma differentiation**. WNT signalling is essential for embryonic development and cell fate specification by executing various programmes[43]. We therefore studied whether depletion of PCF11 'translates' into functional alterations of the WNT pathway and associated cellular programmes. To this end, we generated stable cell lines expressing an Isopropyl β-D-1-thiogalactopyranoside (IPTG)-inducible shRNA against PCF11 (further details see Methods section). Interestingly, PCF11 depletion abrogates WNT signalling in reporter assays (Fig. 4c), and results in cell cycle retardation, increased rate of apoptosis, reduced cell proliferation (Fig. 4d–f, Supplementary Figure 6b) and ultimately differentiation of neuroblastoma (Fig. 4g). Importantly, this effect is not limited to the BE(2)-C cell model and similar results were obtained in CHP-134 neuroblastoma cells (Supplementary Figure 6c). Thus, PCF11 depletion mimics an ATRA-induced neurodifferentiation phenotype, which is associated with the activation of essential pathways for neuroblastoma differentiation

(Fig. 4a)[42]. Vice versa, constitutive overexpression of PCF11 inhibits ATRA-induced neurodifferentiation (Supplementary Figure 6d).

Based on these observations linking PCF11 to hallmark features of cancer, we hypothesised that PCF11 may determine a malignant phenotype. In order to assess this further, we used stable cell lines expressing an IPTG-inducible shRNA against PCF11 (see above). Indeed, depletion of PCF11 abolished colony formation, reduced cell invasiveness and resulted in retarded tumour growth in a neuroblastoma xenograft model (Fig. 5a–c). This recapitulates our findings obtained with the BE(2)-C and CHP-134 neuroblastoma models (Fig. 4d–g, Supplementary Figure 6c) and corroborates an important role of PCF11 in tumour fate specification.

Neuroblastomas originate from incompletely committed sympathetic neural precursor cells. We thus reasoned that PCF11 expression may specify distinct developmental stages. Mirroring PCF11 downregulation during neuronal differentiation (Fig. 3a), we also observed significantly higher PCF11 expression prenatally compared to postnatal human and murine brain samples (p-value $< 8.1 \times 10^{-17}$; Supplementary Figure 7a,b). Accordingly, mature brain tissues show a significant TREND-lengthening phenotype compared to embryonic stem (ES) cells (Supplementary Figure 7c) including all four representative transcripts from the APA-affected module with a role in neurodifferentiation (GNB1, AES, IGF1R and EIF2S1; Fig. 4b). Thus, although neuroblastomas derive from sympathetic nervous system precursor cells, it appears that they share neurodevelopmental features with neurons in the central nervous system with PCF11-dependent APA regulation being an important mechanism in this process.

To further corroborate the role of PCF11 for APA and neurodifferentiation, we generated a transgenic TET-inducible PCF11-shRNA mouse model (further details see Methods section). Briefly, in this model system doxycycline supplementation induces the expression of a shRNA designed to specifically ablate PCF11 expression (Fig. 5d, upper panel). Using this system, we observed APA with a predominating transcript lengthening phenotype upon PCF11 depletion in ES cells and, to a lesser extent, as expected, in mature brain samples (Fig. 5d, lower panel). Strikingly, PCF11 depletion in primary murine neurons (E18) obtained from the central nervous system of these animals led to neurodifferentiation (Fig. 5e), which is consistent with the neurodifferentiation phenotype upon depletion of PCF11 in the BE(2)-C and CHP-134 model system (Fig. 4g, Supplementary Figure 6c). Thus, although BE(2)-C and CHP-134 represent only one of the three genetic subgroups of neuroblastoma (i.e. high-risk MYCN amplified tumours), our data indicate that this phenomenon is not confined to MYCN amplified cells nor

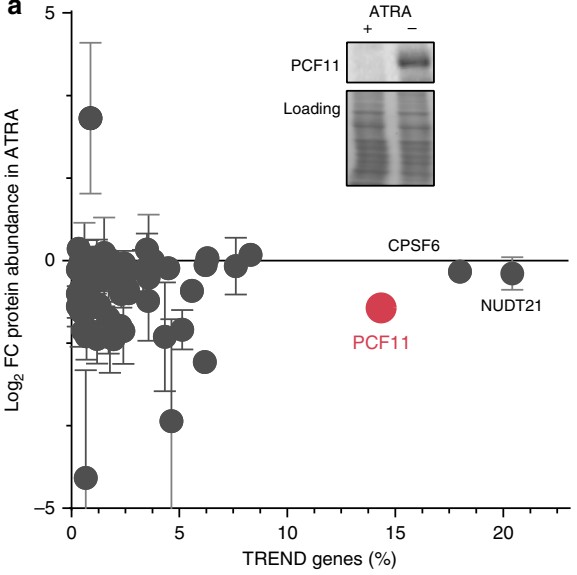

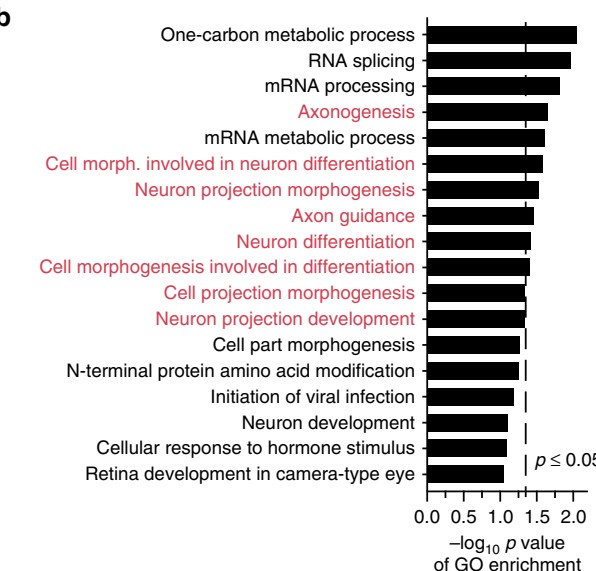

**Fig. 3** PCF11 is a key driver of TREND in neuroblastoma with a potential role in neurodifferentiation. **a** Protein profiling of BE(2)-C cells by western blotting (Supplementary Figure 4) reveals downregulation of PCF11 during ATRA-induced neuroblastoma differentiation (highlighted are the top three TREND-regulators; y-axis, fold-change protein abundance; x-axis, percentage of TREND-affected genes; mean ± s.e.m., two independent replicates; inlet shows a representative PCF11 western blot with equal loading). **b** PCF11 depletion regulates TREND of target genes with a role in neuronal differentiation (GO enrichment analysis of TREND-affected target genes with a protein change upon PCF11 depletion (obtained by differential mass spectrometry of BE(2)-C cells with and without depletion of PCF11) Supplementary Table 4)

restricted to neuroblastoma. This suggests a more global role of PCF11 in coordinating the timely switch to fully committed neuronal fate (Figs. 4g and 5e) thereby preventing uncontrolled embryonic proliferative programmes that may eventually give rise to neuroblastic tumours.

**PCF11 links APA to spontaneous tumour regression**. We next explored whether these findings obtained with various cell models can be generalised and 'translate' into a clinical phenotype. To this end, we made use of a previously thoroughly validated cohort

of 498 neuroblastoma samples[44]. In this data set we observed a significantly more adverse outcome in patients showing high-level compared to low-level PCF11 expression (BH-adjusted p-value: 0.00275; Fig. 5f). In light of PCF11 controlling APA of a module of transcripts associated with neurodifferentiation (Fig. 3b), we speculated that the expression of PCF11 may be associated with spontaneous neuroblastoma regression. Although metastatic, stage 4S neuroblastomas diagnosed in children <12 months show an excellent prognosis (approximately 90% survival) even without chemotherapy due to a high rate of spontaneous tumour regression. In contrast, stage 4 neuroblastomas displaying an almost comparable metastatic spread rarely regress and have a poor outcome (30–50% survival) despite intensive multimodal therapy[45]. When comparing stage 4S with stage 4 neuroblastomas, we identified a significantly higher PCF11 expression in the latter (p-value = $1.15 \times 10^{-6}$, Fig. 5g). Furthermore, low-level PCF11 expression also appears to be unique for stage 4S when compared to the localised stage 1 or 2 (that normally do not spontaneously regress; Supplementary Figure 7d). Thus, low-level PCF11 is associated with a better outcome and possibly a greater likelihood for spontaneous tumour regression. This reflects our in vitro and in vivo observations (Figs. 4c–g and 5a–e, Supplementary Figure 6a–e) and is corroborated by the respective expression signature of established markers for spontaneous neuroblastoma regression (that is a higher expression of HOXC9[46] and CHD5[47], Fig. 5g), although further studies are required to more comprehensively illuminate the role of PCF11 in this process.

In view of the central role of PCF11 for neurodifferentiation (Figs. 4g and 5e, Supplementary Figure 6c), for the regulation of APA of a neurodifferentiation operon (Fig. 3b) and the association with neuroblastoma prognosis (Fig. 5f, including high-risk neuroblastomas, Fig. 6a), we expected to identify corresponding TREND alterations in neuroblastoma patients, which mirror this functional association. To this end, we programmed a bioinformatic pipeline suited to extract transcript isoforms from conventional mRNA expression array data and queried the fraction and identity of transcripts differing at the 3′ end. Indeed, we identified a high prevalence of transcript lengthening (332 genes vs. 93 genes showing transcript shortening) in conditions with low- (stage 4S) compared to high-level PCF11 expression (stage 4; Fig. 6b). This is reminiscent to the TREND phenotype showing mostly a switch to longer transcript isoforms after PCF11 depletion (in vitro and in vivo, Figs. 2b, c and 5d). We further identified a substantial fraction of APA-affected genes (n = 74) that are shared in the clinical cohort (stage 4S vs. 4; Fig. 6b) with those regulated upon PCF11 depletion in BE(2)-C cells (hyper-geometric test of enrichment p-value = 0.004; Supplementary Table 5). Surprisingly, this also extends to 17 out of 26 detectable APA-affected genes belonging to the neurodifferentiation module identified in the PCF11-depletion setup (Figs. 3b and 4a), including AES, IGF1R and GNB1 (Fig. 6b). Specifically, the relative lengthening of the transcript isoforms encoding GNB1, a critical modulator of various transmembrane signalling pathways[48], is significantly associated with a better clinical outcome in neuroblastoma patients (Fig. 6c). Thus, in line with our observations in the reductionist BE(2)-C cell model system (Figs. 2–4 and 5a–c), this finding corroborates the dominant nature of PCF11 in regulating APA in neuroblastoma specimens (compared to other potential APA regulators (Figs. 2 and 3)). It also suggests a potential oncogenic function of deregulated APA in this tumour entity.

**APA of GNB1 is central to PCF11-driven neurodifferentiation**. 3′end diversification can have profound physiological effects by

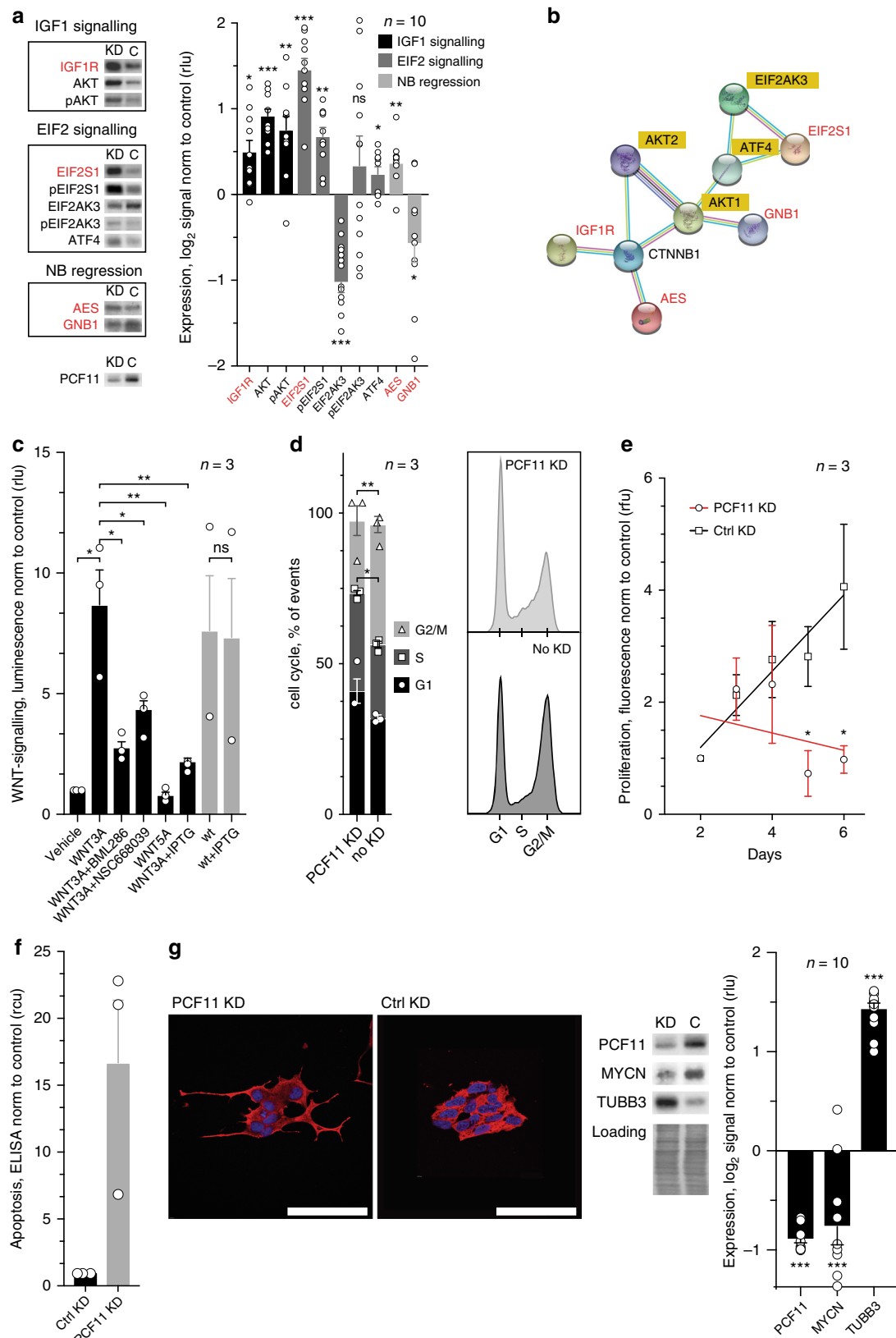

resulting in mRNAs encoding proteins with distinct functions or regulatory properties or by affecting the mRNA fate via the inclusion or exclusion of regulatory elements[6]. However, expression modulation of key factors involved in RNA metabolism, such as PCF11, can have pleiotropic and/or indirect effects

(unrelated to APA) that may contribute to the phenotypes observed in this study. To further substantiate the direct role of PCF11-directed APA for neurodifferentiation, we explored the function of GNB1 3′end transcript isoforms in this process. To this end, we used luciferase reporter assays allowing to study the

**Fig. 4** PCF11-directed APA regulation converges on WNT signalling and modulates cell cycle progression, proliferation, apoptosis and neuronal differentiation. **a** Effect of PCF11-mediated APA regulation on protein expression of APA-affected targets (red) and downstream signalling pathways compared to a control knockdown. **b** Protein-protein interaction network of validated APA-affected candidates (red) impinging on WNT signalling (i.e. CTNNB1, bold; String-DB). This corresponds to the predicted function of the entire cohort of APA-affected genes with a role in neurodifferentiation (compare Supplementary Figure 6a). **c** shRNA-mediated depletion of PCF11 induced by IPTG (see Methods section) inhibits canonical WNT signalling in reporter assays (compare bar 2 with 6). WNT antagonists BML286 and NSC668039, and WNT5A (a non-canonical WNT-ligand) confirm specificity of pathway activation and functionality of the beta-catenin TCF/LEF-driven gene-reporter construct, respectively (for PCF11-APA-directed regulation of WNT signalling and its specific control via different transcript isoforms encoding GNB1 see endogenous TCF7 expression in Fig. 6e). PCF11 depletion results in **d** cell cycle retardation, **e** reduced proliferation, **f** increased apoptosis (ELISA-DNA fragmentation assay; $n = 3$; a loss-of-apoptosis phenotype upon PCF11 overexpression is shown in Supplementary Figure 6b) and **g** triggers neuroblastoma differentiation morphologically and molecularly (for PCF11 depletion-induced neuronal differentiation of neuroblastic CHP-134 cells see Supplementary Figure 6c). Error bars in **a–g** show mean ± s.e.m. of at least three independent experiments, one-sided $t$-test $p$-value; $*p < 0.05$, $**p < 0.01$, $***p < 0.001$; FC fold change, cells are stained with an antibody directed against TUBB3 (red), nuclei are stained with DAPI (blue)

effect of the long and short GNB1 3′ UTR on protein output (Fig. 6d). Consistent with GNB1 transcript lengthening and lower GNB1 protein expression following PCF11 downregulation (Fig. 4a), the long GNB1 transcript isoform appears to harbour a repressive element residing in the long 3′ UTR that reduces GNB1 protein expression (Fig. 6d). In contrast, the short isoform is more efficiently translated, which corresponds to high-level GNB1 protein expression in the presence of PCF11 (Fig. 4a).

We next performed combinatorial depletion experiments to selectively manipulate the abundance of GNB1 transcript isoforms and to disentangle their function. To this end, we used a stable BE(2)-C line expressing an IPTG-inducible shRNA against PCF11 (Fig. 5a–c) and specifically depleted the long GNB1 transcript isoform under conditions with and without IPTG supplementation (Fig. 6e). While PCF11 depletion induced GNB1 transcript lengthening, downmodulation of GNB1 protein and neurodifferentiation (Fig. 6e, f), co-depletion of the long GNB1 transcript isoform restored GNB1 protein abundance, and, surprisingly, antagonised neurodifferentiation induced by PCF11 depletion (Fig. 6e, f). Further, in line with our preceding observations demonstrating a functional role in WNT signalling (Fig. 4b, c, Supplementary Figure 6a) these manipulations are associated with corresponding alterations of the WNT pathway (Fig. 6e, reflected by the regulation of TCF7). Thus, selectively shifting the relative proportion of the long vs. the short GNB1 transcript isoform towards a 'PCF11-high-level' phenotype (that is a more prevalent short isoform) reverts a neurodifferentiation programme induced by PCF11-downmodulation. This directly corroborates the functional importance of PCF11-directed APA regulation, in which alterations of GNB1 transcript isoforms appear to play a central and causative role. It provides a mechanistic explanation for the observed clinical phenotypes, showing a particularly poor patient outcome under conditions of high-level PCF11 expression, predominant expression of short TREND isoforms of a neurodifferentiation operon (including AES, IGF1R and GNB1), followed by downstream aberrations of WNT signalling and defective terminal differentiation of neuroblastomas.

**PCF11-directed TREND signatures are potent biomarkers.** To investigate the clinical implications of these findings, we finally explored whether the functional importance of PCF11-directed APA for neurodifferentiation is more generally reflected in patient outcome, and whether perturbed TREND signatures might thus have prognostic potential. Applying receiver operating characteristic (ROC) curve analysis reflecting the relative abundance of PCF11-regulated long and short transcript isoforms, we identify that individual PCF11-mediated TREND signatures (including GNB1) appear to discriminate high- and low-risk neuroblastomas as well as death surprisingly better than common

clinically used predictive markers of tumour progression (e.g. MYCN amplification; Fig. 7a, Supplementary Table 6, for $p$-value comparison between predictive power of established risk marker expression and combined TREND patterns see Supplementary Table 7, Cox modelling Supplementary Figure 7d). In contrast, the mere RNA abundance of these APA-affected candidates fails to be predictive and shows a comparatively poor prognostic power (Fig. 7b). Of note, this highly prognostic module also includes GNG2, a functionally important dimerisation partner of GNB1.

Altogether, these data illustrate the functional role and prognostic importance of perturbations at the mRNA 3′ end of an APA-affected set of transcripts with a role in neurodifferentiation in neuroblastoma. They establish PCF11 as a critical regulator of neurogenesis with aberrant PCF11 causing pervasive (de)regulation of APA bearing detrimental consequences for normal neurodifferentiation and a functional role in neuroblastoma tumorigenesis. Intriguingly, alterations of the PCF11-GNB1 axis with downstream perturbations of WNT signalling seem to play a central role in this process.

## Discussion

Diversification at the transcriptome 3′ end is a widespread and evolutionarily conserved gene regulatory mechanism. Although TREND has been associated with differentiation and dedifferentiation processes[16,17], the underlying mechanisms and functional consequences are still poorly defined[5,19]. Here we identify massive alterations of the transcriptome 3′end architecture in neuroblastoma. This tumour entity is characterised by a general paucity of somatic mutations[2] and a comparatively high, yet mechanistically enigmatic, incidence of spontaneous tumour regression[45]. By applying a global RNAi screening of a rationally chosen set of factors controlling various facets of RNA metabolism, we delineate the dynamic landscape of TREND in neuroblastoma. While depletion of most TREND regulators leads to an even distribution of transcript isoform shortening and lengthening, the depletion of CFIm and CFIIm complex components belonging to the pre-mRNA 3′end processing machinery results in an almost uniform shortening or lengthening phenotype, respectively (TREND-DB: http://shiny.imbei.uni-mainz.de:3838/trend-db). This suggests an important regulatory function of these TREND regulators in global transcriptome 3′end organisation (with individual factors regulating 1000 and more genes, Supplementary Table 2). Interestingly both, TREND regulators and TREND-affected genes, share high phylogenetic conservation, and numerous components of the canonical 3′end processing machinery control RNA 3′end diversification of genes with an important role in tumorigenesis.

Mechanistically, PCF11, together with CLP1 (both of which form a functional heterodimer), constitute the only TREND

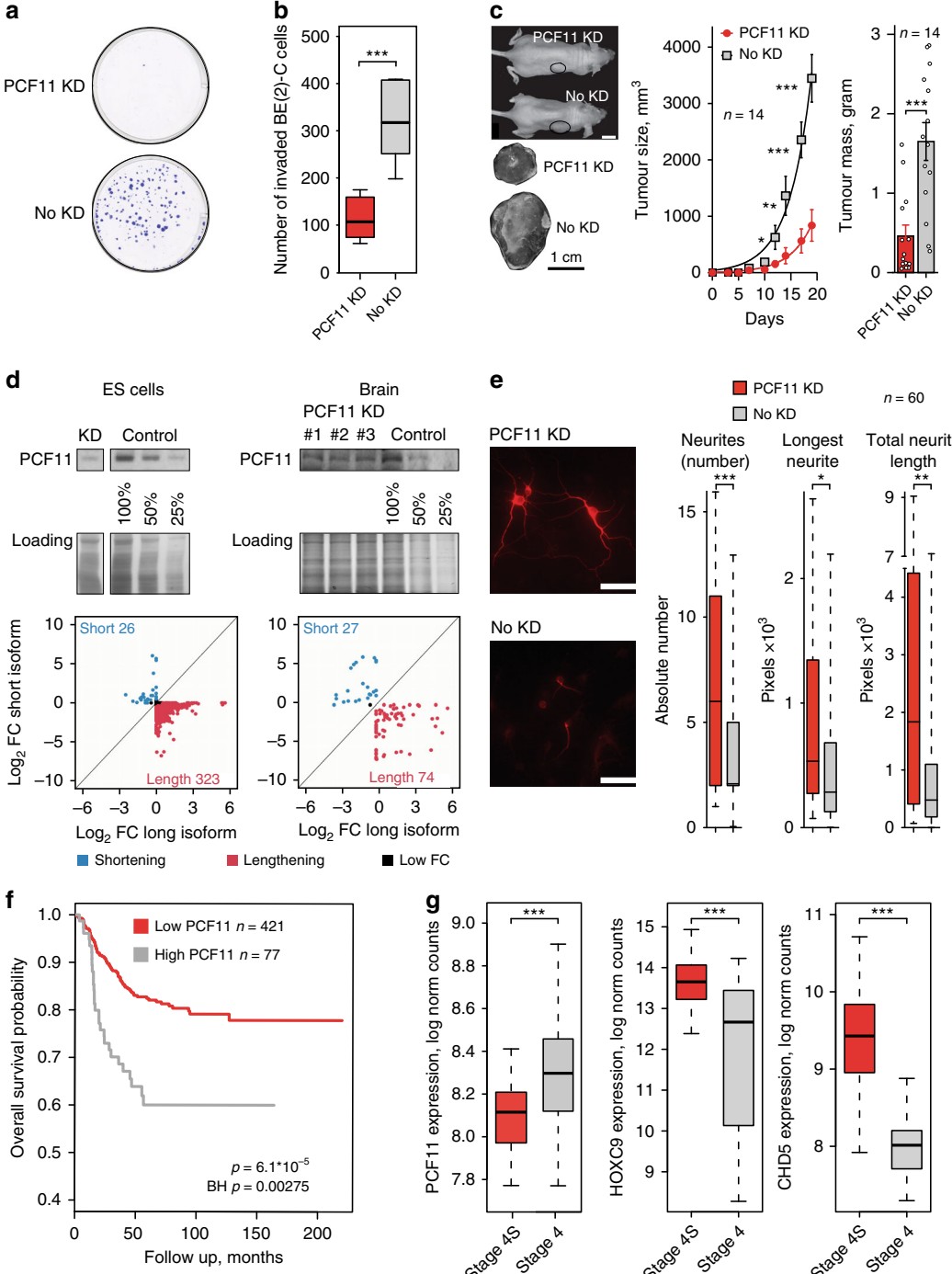

**Fig. 5** PCF11 orchestrates neuroblastoma progression and neuronal differentiation in vitro and in vivo in mice and neuroblastoma patients. **a** Colony formation and **b** matrigel invasion assays of BE(2)-C cells with and without depletion of PCF11. **c** Neuroblastoma xenograft transplantation model with and without depletion of PCF11 (mean ± s.e.m., one-sided $t$-test $p$-value; scale bars 1 cm). **d** In vivo effect of PCF11 depletion on TREND in murine ES cells and in neuronal tissues obtained from a mouse model transgenic for a TET-inducible PCF11-shRNA. **e** Differentiation phenotype of murine neuronal precursors (E18) with and without doxycycline-induced depletion of PCF11 (scale bar 100 μm; cells are stained with an antibody directed against TUBB3 (red)). **f** Kaplan-Meier curve showing low PCF11 expression being associated with higher survival rate in neuroblastomas (BH-adjusted $p$-value, log-rank test, GEO GSE49711[44]). **g** Low-level PCF11 expression in patients with spontaneously regressing stage 4S ($n = 48$) compared to fatal stage 4 ($n = 65$) neuroblastoma tumours. High-level expression of neuroblastoma regression markers in stage 4S ($n = 48$) vs. stage 4 ($n = 65$) neuroblastoma tumours (HOXC9[46] $p = 2.3 \times 10^{-8}$ and CHD5[47] $p = 2.8 \times 10^{-18}$), confirming validity of patient cohorts (two-sided $t$-test $p$-values; GSE49710[44], for comparison of PCF11 expression in spontaneously regressing stage 4S with low-risk neuroblastoma tumours (i.e. stage 1 and 2) see Supplementary Figure 7d). For box plots, centre line depicts median, hinges show 25th and 75th percentile, whiskers depict interquartile range (IQR = 1.5). *$p < 0.05$, **$p < 0.01$, ***$p < 0.001$

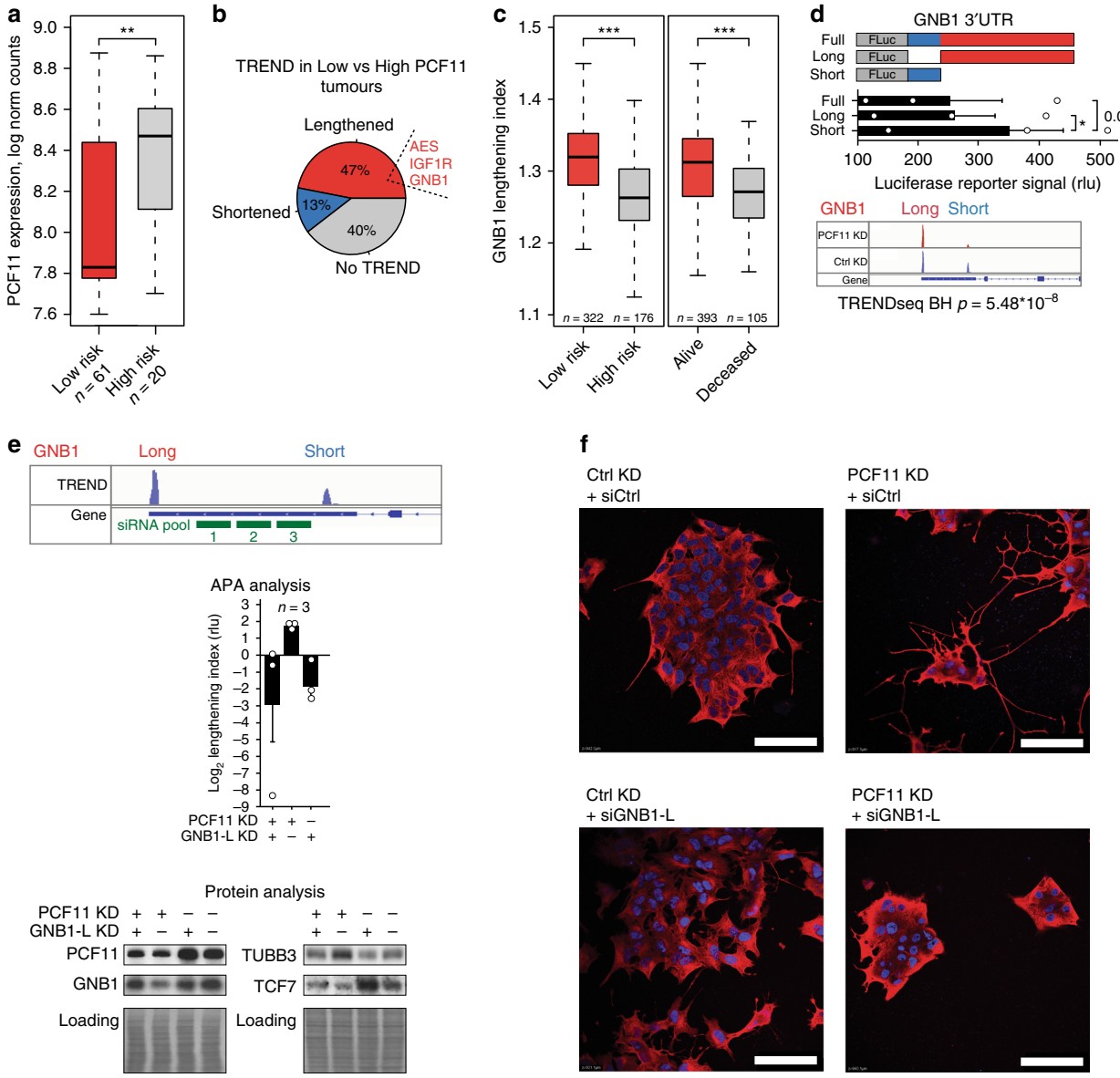

**Fig. 6** Lengthening of the $G_{\beta\gamma}$-complex component GNB1 transcript is a central checkpoint in PCF11-TREND-mediated neuroblastoma differentiation. **a** PCF11 is significantly lower in low-risk compared to high-risk neuroblastoma patients ($p = 0.005172$; two-sided $t$-test; GSE49710[44], ordered according to PCF11 expression; 10th percentile). **b** Low PCF11 in stage 4S neuroblastoma patients (compare Fig. 5g) is associated with pervasive TREND-lengthening (Supplementary Table 5; stage 4S, $n = 48$; stage 4, $n = 65$) including AES, IGF1R and GNB1. **c** GNB1 transcript lengthening correlates with a superior outcome in neuroblastoma patients (two-sided $t$-test). **d** GNB1 protein abundance (recorded by luciferase reporter assay) is determined by the GNB1 3′ UTR transcript isoform (middle; scheme of GNB1 3′ UTR luciferase reporter is shown in top). 'Full' and 'Short' denote 3′ UTR luciferase reporter constructs harbouring the (full-length) GNB1 3′ UTR processed at the distal or proximal polyadenylation signal (compare IGV track on the bottom before and after PCF11 depletion, for details see also legend to Fig. 2a). 'Long' indicates a construct with a 3′ UTR corresponding to sequences unique for the long transcript isoform (bar diagrams represent mean ± s.e.m. and a one-sided $t$-test $p$-value). **e** Depletion of the long GNB1 transcript isoform (siGNB1-L) in a low- or high-level PCF11 background (siRNA target sites highlighted in IGV tracks on top) results in reciprocal GNB1 mRNA isoform expression (middle panel) and protein output (lower panel) and counteracts a PCF11-KD-driven inhibition of WNT signalling (TCF7, compare Fig. 4c) and differentiation of neuroblastoma molecularly (TUBB3) and **f** phenotypically (cells are stained with an antibody directed against TUBB3 (red), nuclei are stained with DAPI (blue), scale bar $= 100\,\mu m$). For box plots, centre line depicts median, hinges show 25th and 75th percentile, whiskers depict interquartile range (IQR = 1.5). *$p < 0.05$, **$p < 0.01$, ***$p < 0.001$

regulators whose downregulation predominantly results in APA far downstream (Fig. 2b, c, Supplementary Figure 2d)[49]. This is consistent with the role of PCF11 in modulating Pol II processivity and transcription termination[50], and thereby provides a possible mechanistic explanation for the widespread lengthening of 3′ UTRs in the mammalian brain[39]. Neuronal PCF11 expression drops around birth and during neuronal differentiation, but appears to be high in neuroblastomas and, interestingly, other

paediatric cancer entities with embryonic origin (Fig. 7c). Thus, it is tempting to speculate that sustained (postnatal) PCF11 expression may drive highly proliferative embryonic programmes by arresting cells in an undifferentiated state. This could also explain the high frequency of microscopic neuroblastic lesions in fetuses[51] or young infants[52] compared to older ages, although future studies are needed to dissect the role of PCF11 for embryonic development in further detail.

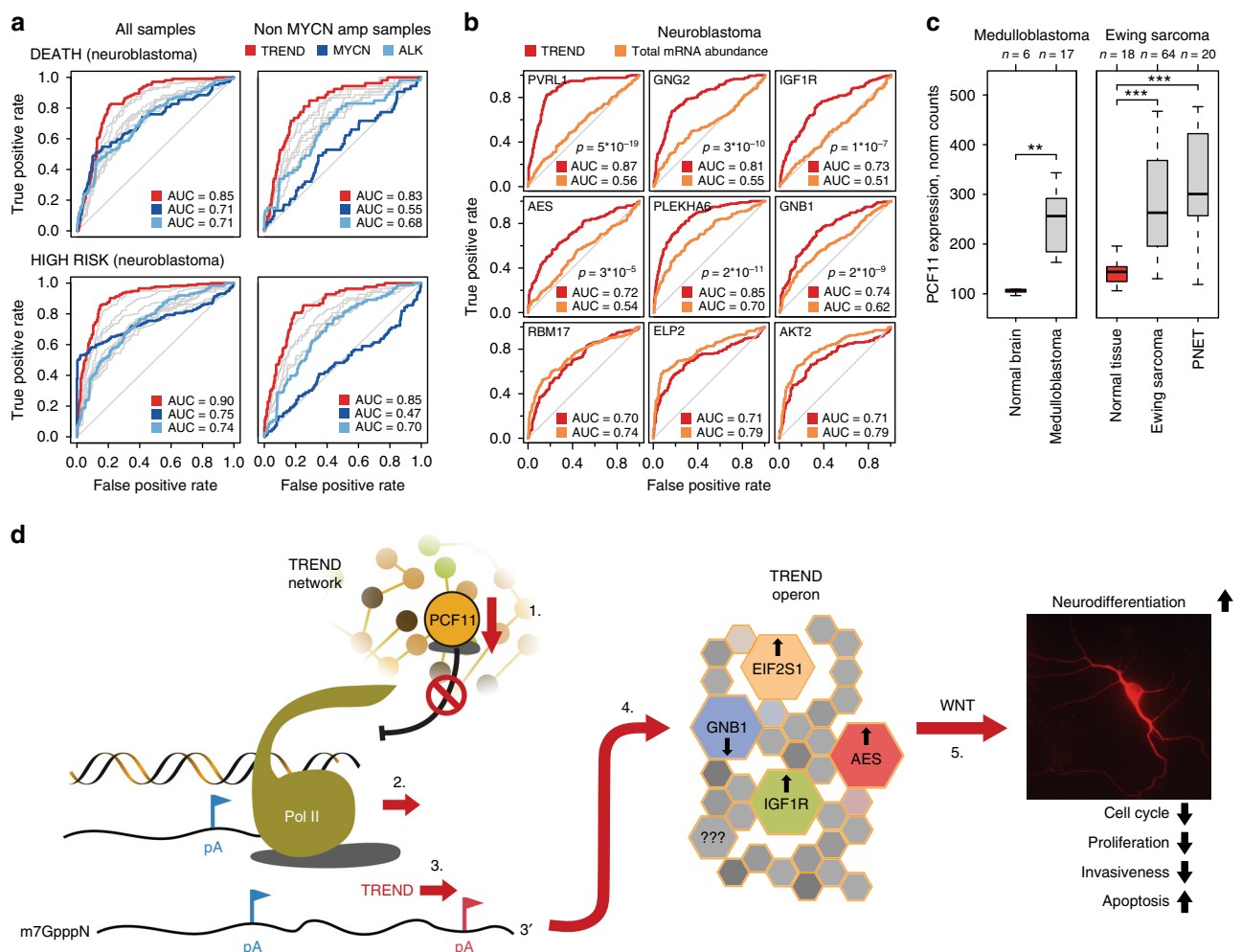

**Fig. 7** PCF11-derived TREND signatures predict superior patient outcome. **a** Predictive potential of TREND signatures (red) compared to common stratifiers in neuroblastoma (blue, Supplementary Table 6) with ($n = 493$, left) or without ($n = 401$, right) MYCN amplification ($p$-values, Supplementary Table 7, Cox-modelling Supplementary Figure 7e). Receiver-operating characteristic (ROC) curve reflecting the ratio of short-to-long isoform abundance of PCF11-TREND-affected genes (with AUC > 0.7) for predicting death (upper panels) or association with high risk (lower panels). Red line depicts predictive power of a combined TREND pattern (multifactor ROC) compared to ROC based on expression of established risk markers (MYCN and ALK[82]; grey lines reflect ROC for 3′end isoform patterns of individual TREND-affected genes, GSE49710[44]; Supplementary Table 6). **b** ROC curves reflecting TREND alterations (red) of genes belonging to the neurodifferentiation TREND operon highlight the predictive power of deregulated TREND signatures compared to the poor sensitivity and specificity of the mere mRNA abundance (orange; $n = 493$, raw data same as for **a**; TREND isoforms and mRNA abundance of RBM17, ELP2 and AKT2 (lower row) serve as control confirming absence of analytical bias in favour of TREND). **c** High-level PCF11 mRNA expression in other paediatric malignancies with embryonic origin including medulloblastomas (GSE35493), Ewing sarcomas (GSE17679) and primitive neuroectodermal tumours (PNET, GSE17679) compared to respective controls (red; expression data obtained from R2 (http://r2.amc.nl)). For box plots, centre line depicts median, hinges show 25th and 75th percentile, whiskers depict interquartile range (IQR = 1.5), two-sided $t$-test. **$p < 0.01$, ***$p < 0.001$. **d** Integrated model for PCF11-dependent TREND regulation in neuroblastoma governing neurodifferentiation (details see text). Briefly, (1) postnatal downregulation of PCF11 is believed to (2) reduce efficiency of transcription termination[20,21], facilitating (3) alternative polyadenylation (APA) at distal (3′ located) polyadenylation (pA) signals (red), affecting TREND of >900 RNAs (4). This includes a neurodifferentiation TREND operon (Fig. 3b), shaping (5) WNT signalling to induce neurodifferentiation. Conversely, sustained (postnatal) high-level PCF11 expression arrests neuronal precursors in an undifferentiated state by promoting polyadenylation at proximal pA (blue), thereby controlling protein output of the neurodifferentiation TREND operon. Transcript isoform regulation of GNB1, a modulator of various transmembrane signalling pathways[48] with a role in development[63], malignant transformation[64] and neurodevelopmental disorders[65] is prerequisite for PCF11-dependent regulation of neurodifferentiation. Notably, GNB1 was previously identified as marker for spontaneous neuroblastoma regression[41]

This study puts PCF11 at centre stage as it affects TREND in neuroblastoma most quantitatively and in a dominant manner. PCF11-directed APA controls a neurodifferentiation RNA operon. Downregulation of PCF11 induces neuronal differentiation of BE(2)-C and CHP-134 neuroblastomas and murine neuronal precursors, conversely forced overexpression halts neurodifferentiation. This suggests a more general critical function of PCF11 and the resulting downstream organisation of

TREND in this process. In line with these findings, we identify (de)regulated APA as a pivotal mechanism linking aberrant PCF11 expression with neuroblastoma formation and prognosis; high-level PCF11 specifies a poor prognosis ($p = 0.00275$) and a low likelihood of spontaneous tumour regression ($p < 0.001$).

Our data suggest the WNT signalling pathway to represent an important convergence point for neurodifferentiation, which, when deregulated through aberrations of APA, contributes to

neuroblastoma genesis. Interestingly, aberrant WNT signalling has been implicated in neuroblastoma formation[53–59]. Among APA-affected transcripts, the $G_{\beta\gamma}$-complex component GNB1 appears to have a decisive function in this process. In addition to its function in heterotrimeric G-protein signalling with a role in pathways essential to development, it modulates WNT signalling (Figs. 4c and 6e)[60–62], and selective depletion of its long transcript isoform abolishes an otherwise dominantly driven neurodifferentiation phenotype induced by PCF11 downregulation. Notably, GNB1 was previously identified as a marker for spontaneous neuroblastoma regression[41]. GNB1 is also affected by APA in a PCF11-like manner upon depletion of BARD1 (see TREND-DB), one of the few factors in which germline mutations have been linked to neuroblastoma formation[13]. Apart from its global role in development[63], constitutive activation of GNB1 has been shown to promote malignant transformation[64] and neurodevelopmental disorders[65]. Thus, together with IGF1R[66], AES[67,68] and EIF2S1 (at least) four members of the neurodifferentiation APA-RNA operon converge on WNT signalling (Fig. 4b, c, Supplementary Figure 6a) with important functional implications (Figs. 4c–g, 5a–e and 6e, f). This reflects the critical role of WNT in directing cell fates during embryogenesis and in a variety of human cancers[69]. Aberrant PCF11, therefore, could functionally 'compensate' for lack of germline mutations in WNT signalling molecules of (hitherto unidentified) WNT-driven pathologies. This could account for neuroblastomas, and may also explain why many cancers without mutations in the WNT pathway still rely on aberrant WNT signalling for proliferation or survival[70], offering an interesting and to be further explored possible mechanism for the effect of APA on neurodifferentiation and beyond.

Post-transcriptional perturbances are increasingly recognised to play a critical role in tumorigenesis, including childhood neuroblastomas[71]. The 3′ UTR aberrations identified here in neuroblastoma can be easily dismissed as irrelevant and/or remain undetected as they affect non-coding elements. Our data highlight the functional relevance of these non-coding elements in driving most devastating processes. They also illustrate that functionally most relevant alterations at the transcriptome 3′ end are prone to escape conventional gene expression profiling[19]. Our observations suggest that TREND signatures could represent powerful biomarkers for neuroblastoma risk stratification and may have the potential to explain previously counterintuitive (clinical and basic research) findings. However, comprehensive studies based on larger and independent patient cohorts are required to further illuminate the clinical implications. In addition, the identification and analysis of aberrations of the transcriptome 3′ end is likely to have far-reaching implications for the elucidation of disease mechanisms beyond neuroblastoma formation[11] and the quest for novel tailored therapies[72].

## Methods

**siRNA library**. A custom siGENOME library was purchased at Dharmacon (GE Healthcare) and consisted of siRNA smartpools to deplete targets listed in Supplementary Table 2.

**Mammalian cell lines**. Neuroblastoma cell lines (CHP-134, LAN-6 and SH-SY5Y) were purchased from Leibniz Institute DSMZ—German Collection of Microorganisms and Cell Culture. The BE(2)-C cell line was a generous gift from Prof. Olaf Witt (DKFZ, Heidelberg). All cells were tested negative for mycoplasma, and propagated in monolayer culture in Dulbecco's modified Eagle's medium medium with 10% foetal bovine serum (10–30 passages). BE(2)-C were differentiated using 5 μM ATRA.

**Transfection of mammalian cell culture**. Cells were plated (60–70% confluence) 12 h prior to the transfection procedure. A master mix for transfection included RotiFect RNAi Lipo (Roth, 2 μl/1 ml of total transfection volume), siRNA (50 nM final) or plasmid (0.5 μg/ml final) mixed in OptiMEM (ThermoFisher Scientific)

in 20% of final volume according to the manufacturer's instruction. Cells were assayed 48–72 h after transfection.

**Generation of stable cell lines**. To generate clones with IPTG-inducible expression of specific shRNA directed against PCF11 or firefly luciferase (as negative control respectively), BE(2)-C and CHP-134 were transfected with pLKO-puro-IPTG-3xLacO constructs (Sigma-Aldrich). Antibiotic selection was performed with 3 μg/ml of puromycin (ThermoFisher Scientific).

Stable cell lines overexpressing PCF11 were generated by transfection of the wild-type BE(2)-C and CHP-134 neuroblastoma. For stable overexpression, a pCI-neo plasmid containing a full-length PCF11 coding sequence with N-terminal Flag peptide was transfected. An empty pCI-neo vector was used to generate a control cell line. For selection of clones with stable integration, Geneticin® G-418 (ThermoFisher Scientific) treatment was initiated 48 h after transfection (1 mg/ml) and carried out for the following 10 days. Individual clones were then propagated and tested for target protein overexpression or depletion using western blotting[73].

**Constructs**. For WNT-luciferase reporter assays, the plasmid M50 Super 8x TOPFlash (#12456, AddGene) was used. It contains TCF/LEF sites for beta-catenin-mediated transcriptional activation upstream of a firefly luciferase gene. Co-transfection was performed with pRL-TK (#E2241, Promega), and firefly luciferase luminescence was normalised to renilla luciferase.

For GNB1 3′ UTR reporter assays, the plasmid pmirGLO (#E1330) was used. Complete or parts of GNB1 gene 3′ UTR were cloned downstream of the Firefly luciferase open reading frame using the following primers:
FF_short: ATACAAGCTAGCCGCCAGTAGCATGTGGATGC;
Rev_short: GATGGCCTCGAGTCAAGTTTACCTTCTGGTTA;
FF_long: ATACAAGCTAGCGTAAACTTGAGTGTAATTGT;
Rev_long: GATGGCCTCGAGGTCCCTCATGTCAAACTGCT

A set of constructs containing specific shRNA sequences to target human PCF11 mRNA was designed by Sigma-Aldrich based on pLKO-puro-IPTG-3xLacO backbone. Target sequences for PCF11 depletion were:
ATCGAAATCGAAATCGAAATC, AGTAGCCTCCCACTGATTAAA, AGATCCTGCTTGGCCTATTAA (was used throughout all functional experiments), CAATCAGACTGGTCCATATAA and TTTGCCATCGGTCTTATC.

For overexpression studies, the coding sequence of human PCF11 protein was cloned into pCI-neo vector backbone (#E1841, Promega). The cDNA was cloned by fusion of two fragments. Amplicons were synthesised using the two following primer pairs for the 5′ and 3′ fragment respectively:
FF1: AAGCCACCGCTCGAGTCAGAGCAGACGCCGGCC,
Rev1: AAGCCACCGCTCGAGATGTCAGAGCAGACGCCGGCC;
FF2: GTGTGCGAGAAGAGCAGAGA,
Rev2: CGGGTCGACTCTAGATTAAACTGACTCGACTGTGTCAT

Both parts were inserted into pCI-neo with a Flag peptide-coding sequence in the multiple-cloning site. Ligation was performed using the InFusion cloning kit (Clontech® Laboratories) according to the manufacturer's instructions.

**Northern blotting**. Total RNA was extracted from cells using PeqGold TriFast (VWR). Northern blotting analysis (in an agarose gel system) was performed as described previously[73]. For higher resolution (in the range of 200–2000 nt) polyacrylamide gel systems were used. Specifically, total RNA was deadenylated in presence of 200 pmol of oligo-(dT)$_{25}$ and 5 units of RNase H. Purified RNA (1–3 μg) was heat-denatured in 50% formamide in TAE and separated in a polyacrylamide gel consisting of 3.5% polymerised acrylamide-methylenebisacrylamide mix (37.5:1), 7 M urea. Electrophoresis was performed in TAE buffer for 4 h at 140 V. RNA was transferred to a nylon membrane (ThermoFisher Scientific) by semi-dry electroblotting. Membranes were ultraviolet cross-linked and hybridised with specific probes in analogy to the standard procedure[73].

**Non-radioactive DNA probe synthesis and detection**. For specific DNA probe synthesis (for Northern blotting), RNA was reversely transcribed. PCR amplicons (of about 450–550 base pairs long) were generated using OneTaq Master Mix (New England BioLabs). Purified PCR products (1–3 ng) were used in the next round of asymmetric PCR (reverse to forward primers ratio 100:1) to generate biotin-labelled probes (50 PCR cycles). Probes were purified by extraction from agarose gels by using NucleoSpin® Gel and PCR Clean-up kit (Macherey-Nagel). For hybridisation, 50–100 ng of probe was used in 5 mL of Church buffer in analogy to procedures described previously[73]. For the detection of biotinylated probes, membranes were washed and blocked for 1 h in solution containing 1x TBS, 0.5% SDS and 0.1% of Aurora™ Blocking Reagent (#04821548, MP Biomedicals). Streptavidin−Peroxidase Polymer (#S2438, Sigma-Aldrich) was used next (diluted 1/7000) for 1 h, following 3 cycles of 10 min washing (with 1x TBS, 0.5% SDS). Signal detection was carried out by using ECL Select Western Blotting Detection Reagent (GE Healthcare). Blots were scanned using ChemiDoc™ MP System (Bio-Rad) and bands were quantified (Image Lab™ software; Bio-Rad).

**Western blotting**. Protein lysates were generated as described previously[73] and separated using Criterion™ TGX™ Stain-Free™ gel (Bio-Rad Laboratories) followed by transfer onto nitrocellulose membrane (GE Healthcare). Equal sample loading was assayed by in-gel fluorescent detection or Ponceau S staining of the membrane. Membranes were blocked with TBS buffer containing 5% milk for 1 h and probed with specific antibodies. All antibodies used were purchased at Bethyl Laboratories with exception for: IGF1R, EIF3AK, pEIF3AK, ATF4, GNB1 and MYCN (Santa Cruz Biotechnology); AKT and pAKT (Cell signalling Technology); pEIF2S1 (Abcam); TUBB3 (BioLegend); AES (Novus Biologicals). Blots were scanned using ChemiDoc™ MP System (Bio-Rad) and bands were quantified Image Lab™ software (Bio-Rad). Uncropped blots of the main figures are shown in Supplementary Figure 8.

**Neuroblastoma TREND annotation assembly**. 3′READS was carried out as previously described[15]. Briefly, total RNA obtained from differentiated and undifferentiated neuroblastoma cell line (BE(2)-C) was subjected to 1 round of poly(A) selection using oligo(dT) beads (NEB), followed by fragmentation on-bead with RNase III (NEB). Poly(A)-containing RNA fragments were isolated using the MyOne streptavidin C1 beads (Invitrogen) coated with a 5′ biotinylated chimeric dT$_{45}$ U$_5$ oligo (Sigma), followed by washing and elution through digestion of the poly(A) tail with RNase H. The part of the poly(A)-tail annealed to the U residues of the oligo was refractory to digestion and was thus used as evidence of the poly(A) tail. Eluted RNA fragments were purified by phenol-chloroform extraction and ethanol precipitation, followed by sequential ligation to a 5′-adenylated 3′-adapter (5′-rApp/NNNNGATCGTCGGACTGTAGAACTCTGAAC/3ddC) with the truncated T4 RNA ligase II (NEB) and to a 5′ adapter (5′-GUUCAGAGUUCUA CAGUCCGACGAUC) by T4 RNA ligase I (NEB). The resultant RNA was reverse-transcribed with Superscript III (Invitrogen), followed by 12 cycles of PCR amplification with Phusion high fidelity polymerase (NEB). cDNA libraries were sequenced on an Illumina HiSeq 2500.

Filtered poly(A) site-supporting (PASS) reads were used to construct peaks using the Cufflinks software. Obtained peaks were associated with the UCSC assembly of *human* genes based on the peaks position using the closest-features command of BEDOPS toolkit. PASS mapped within the 5000 nucleotide region downstream of the annotated gene were considered as novel 3′ UTR isoforms. This annotation was used to exclude TRENDseq reads originating from internal priming on the genome encoded adenosine-rich regions. For mouse annotation of true poly(A) sites, data (PMID 24072873) were used with settings identical for human neuroblastoma annotation.

**Nucleic acid quality assurance**. RNA integrity was assayed with Agilent RNA 6000 Nano Kit (Agilent Technologies) according to the manufacturer's instructions[74], and a threshold of minimal RNA integrity number of 9.5 was applied for total RNA. Homogeneity and size of DNA libraries for Illumina sequencing were analysed using Agilent High Sensitivity DNA Kit (Agilent Technologies) following the manufacturer's instructions. Qubit® 2.0 Fluorometer in combination with dsDNA HS Assay Kit (ThermoFisher Scientific) was used to assay cDNA library concentration.

**TRENDseq: library preparation and sequencing**. Total RNA (100 ng) was reverse-transcribed in presence of oligonucleotide (RT) primer containing T7 promoter, Illumina 5′ adapter, individual in-lane barcode and an anchored oligo-dT stretch, as described previously[75]. For the cDNA and aRNA synthesis MessageAmp II aRNA Amplification Kit (ThermoFisher Scientific) was used according to the manufacturer's recommendations with modifications. Specifically, for the first and second cDNA strand synthesis for each individual RNA input sample 1/10 of full reaction size was used. Up to 25 samples were pooled after second cDNA strand synthesis reaction. In vitro transcription (aRNA synthesis) was performed in 40 μl reaction format according to the manufacturer's protocol with 14 h of incubation at 37 °C. Purified aRNA was sheared using Covaris M220 Focused-Ultrasonicator™ (Peak incident power 50 Watt, Duty Factor 20% and 200 Cycles per Burst (cbp) for 420 s at 7 °C) and size-selected on the 6% PAGE in denaturing conditions (7 M urea). The gel region corresponding to 100 nucleotides was excised, and RNA was eluted from gel by 2 min incubation in 50–100 μl of buffer 100 mM Tris-HCl (pH 8.0), 500 mM NaCl and 1% SDS at room temperature. Size-selected RNA was purified using miRNeasy Kit (Qiagen).

Illumina platform compatible cDNA library was synthesised as described previously[75] with a number of PCR cycles reduced down to 9. Each pooled library (up to 25 samples) was labelled with Illumina indexing barcode and up to three libraries were pooled together adding up to 75 samples per sequencing run. The libraries were sequenced on the Illumina HiSeq or NextSeq platform with addition of 30% PhiX Sequencing control (Illumina) in the paired-end setup. Read 1 (9 nt) sequenced individual sample in-lane barcode (introduced in the first reverse transcription-step) and read 2 (50 nt) sequenced the RNA insert to be mapped to the genome (Fig. 2a). Illumina indexes were sequenced as a dedicated read.

**TRENDseq: bioinformatical analysis**. Raw sequencing data (fastq format) were demultiplexed using in-lane RT primer barcode with the average per base quality score above 20, followed by A- and T-stretches trimming as described previously[75].

Resulting sequences with average length of 25–35 nucleotides were mapped to human hg38 genome using bowtie2 aligner. Mapped reads were filtered from internal priming events using the assembly of TREND annotation (see above). The number of reads associated with each TREND isoform was calculated using HTSeq (htseq-count command, intersection-strict option). Number of reads aligned to each site reflect expression of an individual 3′end position.

For statistical analysis, the expression level of each transcript isoform was examined by Fisher's exact test in comparison to the respective other alternative 3′end isoform(s) expressed by the same gene. Contingency table included the number of reads of the tested isoform and total amount of reads of all the other isoforms of the gene (for the knockdown and control samples, respectively). Obtained p-values were adjusted using the Benjamini-Hochberg method, and adjusted p-value ≤ 0.05 filter was applied. To calculate fold-regulation per isoform, total amount of reads for each gene was normalised to 100%, and percentage of individual isoform in the knockdown sample was divided by the percentage of the same isoform in the control.

The most 3′end position and fold change of regulated transcript isoforms (Fig. 2b and Supplementary Figure 3c) was calculated relative to the 'Zero-isoform', which is defined as the longest significantly affected (BH-adjusted p ≤ 0.05) and annotated transcript isoform expressed by the respective gene. To describe the overall tendency of a given gene to express shortened (or lengthened, respectively) transcript isoforms, a proxy of two most significantly affected isoforms was applied (Supplementary Figure 2c, Fig. 3b and Supplementary Figure 3b, d). The shortening index was calculated as the fold-regulation of the shorter isoform normalised to the fold-regulation of the longer isoform of the same gene (a positive log2 shortening index represents a higher abundance of the shorter transcript isoform upon depletion of the respective TREND regulator, and vice versa). GO analysis was performed using DAVID functional annotation tool (version 6.7). The network analysis (Supplementary Figure 3b) was built according to the Fruchterman-Reingold algorithm based on force directed nodes placement, wherein the distance between the nodes reflects the total number of affected genes and significance of antagonistic or synergistic action by repulsion or attraction, respectively.

**Programming and packages**. TRENDseq data analysis and visualisation was performed in the R environment[76] (https://www.R-project.org/). Packages used: gplots, igraph, ggplot2, pROC, limma, survival.

**Immunofluorescent micrographs**. Cells were plated onto sterile microscopy coverslips and propagated under the experimental conditions (detailed above). Fixation was performed for 10 min in phosphate-buffered saline (PBS) solution containing 4% paraformaldehyde and 10 mM NH4Cl. Cells were permeabilised with 0.2% Triton X-100 in PBS at room temperature for 10 min. Cells were blocked with 2.5% Normal Horse Serum Blocking Solution (Vector Laboratories) for 1 h and stained with a primary antibody directed against TUBB3 in PBS solution containing 1% bovine serum albumin (BSA) and 0.05% Triton X-100. After staining with Cy3-labelled secondary antibody (anti-rabbit) samples were mount onto microscope slides with Vectashield Antifade Mounting Medium containing the 4′,6-diamidino-2-phenylindole (DAPI) stain (Vector Laboratories). Image analysis was performed using ImageJ.

**Apoptosis assay**. Apoptosis measurements were performed using the Cellular DNA Fragmentation ELISA kit (Sigma-Aldrich). BE(2)-C cells were plated in black 96-well plates with transparent bottom in duplicates (5000 cells/well). A unit of 1 mM IPTG was added to the cells to induce PCF11 depletion. Seventy-two hours after plating, cells were labelled with 10 μM bromodeoxyuridine (BrdU) for 24 h. After BrdU withdrawal, cells were kept in culture for another 48 h. For cell lines with stable PCF11 overexpression, 3 μg/ml of puromycin or 1 μM lometrexol (Sigma-Aldrich) was applied to trigger apoptotic response for the same period after BrdU withdrawal. After 48 h and prior to the harvesting procedure, cells were labelled with NucBlue® Live ReadyProbes® Reagent (ThermoFisher Scientific) for 30 min (2 droplets of reagent per 1 ml of media). After washing with PBS, cells were imaged with fluorescence microplate reader (Fluoroskan Ascent FL, ThermoFisher Scientific), and the signal at the wavelength ~461 nm was used as a measure of cell number in the well (as normalisation control). Thereafter, the ELISA procedure was performed according to the manufacturer's instructions (Cellular DNA Fragmentation ELISA kit protocol). The colorimetric signal was normalised to the cell number in each well.

**Cell cycle analysis**. PCF11 knockdown in BE(2)-C cells was performed by siRNA transfection as described above. Forty-eight hours after transfection, cells were synchronised with 2 mM hydroxyurea (HU) for 16 h. Cells were harvested at 0, 3, 6, 9 and 12 h after HU withdrawal. Cells were treated with 30–50 μg/ml of propidium iodide (ThermoFisher Scientific) solution in 0.1% Triton X-100 in PBS in presence of 2 mg of DNase-free RNase A (ThermoFisher Scientific). After staining for 15 min at 37 °C, the fluorescent signal was measured with a LSR II Flow Cytometer (BD Biosciences) using a 670 nm long-pass filter. Cell doublet discrimination was performed using FSC-H/FSC-A, SSC-H/SSC-A and PI-H/PI-A gates.

**Proliferation assay**. Cells were plated in black 96-well plates with clear bottom in 10 replicates (3000 cells/well). PCF11 knockdown was performed by addition of 1 mM IPTG to the culture medium (see above). After 48 h of incubation, two replicates were stained for 30 min with NucBlue® Live ReadyProbes® Reagent (ThermoFisher Scientific) according to the manufacturer's recommendations. Cells were washed in PBS, and the signal was measured with a fluorescence microplate reader (Fluoroskan Ascent FL, ThermoFisher Scientific, ~460 nm wavelength). The procedure is repeated over 5 days with two fresh replicates to assay proliferation kinetics.

**Colony formation assay**. For colony formation assays, 200 BE(2)-C cells (see above) were plated into each well of a six-well plate for 10 days without and with addition of IPTG. Thereafter, the cells were washed and fixed, and colonies were stained with 0.5% crystal violet for 10 min at room temperature.

**Matrigel invasion assay**. To assess the invasive properties upon PCF11 depletion, 80,000 BE(2)-C cells (see above) were seeded into the insert of a growth factor reduced 24-well Matrigel™ invasion chamber assay plate (BD BioCoat™) without and with addition of IPTG (in 0.5 ml serum-free media); the inserts were transferred into wells containing 0.75 ml culture medium with 10% fetal calf serum with and without IPTG. After 72 h, non-invading cells were removed from the upper surface of the membrane with a cotton swab, and invading cells were fixed and stained with Diff-Quick® (Medion Diagnostics) and quantified by counting invaded cells in four independent areas in pentaplicates. All experiments were performed in accordance with the manufacturer's protocols.

**Mouse tumour xenotransplantation**. Female athymic nude mice (Crl:NU(NCr)-Foxn1nu, Charles River) were used to assess tumour progression of BE(2)-C cells in response to the PCF11 expression status. To that end, $12.5 \times 10^6$ cells each in 0.2 ml PBS were subcutaneously injected into the right/left flank of 30 nude mice (aged 6 weeks, weighting ~20–25 g). For PCF11 depletion, half of the animals were randomly assigned to a cohort (on day 3 after tumour transplantation), which received IPTG injections (1.95 M, Roth, Germany) every second or third day (injection of PBS served as negative control), and tumour dimensions were measured with callipers every 2 or 3 days. Tumour volume was calculated by modified ellipsoid formula (½ × (length × width²)); mice were sacrificed after a follow-up of ~20 days after tumour cell injection and tumours were removed and weighted. All animal experiments were approved by local authorities (Rhineland-Palatinate), and animals' care was in accordance with institutional guidelines.

**Luciferase reporter assay**. For the WNT reporter assays wild-type BE(2)-C or clones with inducible shRNAs directed against PCF11 (see above) were plated 12 h prior to the procedure in 24-well plates with or without 1 mM IPTG. Transfection with 1.6 µg of TOPFlash plasmid and 0.4 µg of pRL-TK (control) was performed in antibiotic-free medium. Modulators of WNT and IGF1R pathways were added in serum-free OptiMEM medium 48 h after reporter transfection. WNT pathway activation was induced by addition of recombinant WNT-3a (canonical) or WNT-5a (non-canonical WNT-ligand; each 250 ng/ml, R&D Systems®, diluted in 0.1% BSA-PBS). WNT pathway inhibition was carried out by addition of 0.2 mM BML-286 (Enzo Life Sciences) or 1 mM NSC668036 (Sigma-Aldrich). IGF1R was inhibited by adding 25–50 µM Tyrphostin AG1024 (Enzo Life Sciences). Dimethyl sulphoxide was used as a solvent control for inhibitors of the WNT and IGF1R pathways.

For monitoring WNT activity, luciferase assays were carried out 24 h after compound addition (see above). Cells were lysed in 12-well plates with Passive Lysis buffer (Promega) for 15 min at room temperature. Firefly and Renilla luciferase luminescence was assayed in reactions with Bright-Glo™ reagent (Promega) and coelenterazine (Promega), respectively. Firefly luciferase was used as a read-out for WNT signalling pathway activation and normalised to the luminescence of Renilla luciferase (vector delivery control).

For analysis of the effect of GNB1 3′ UTR isoforms on luciferase expression, pmirGLO-constructs were transfected into wild-type BE(2)-C in 12-well plates. Luciferase activity was assayed 24 h later as described above.

**Inducible PCF11-RNAi mouse model**. For reversible depletion of PCF11, an inducible knockdown allele of the PCF11 gene was generated via targeted transgenesis of a doxycycline-inducible shRNA cassette into the ROSA26 locus (Gt (ROSA)26Sor)[77]. Briefly, to that end a recombination-mediated cassette exchange vector harbouring an inducible H1 promoter (H1tetO)-driven shRNA cassette along with a genetic element for the constitutive expression of the codon-optimised tetracycline repressor protein (iTetR), and a neomycin resistance cassette was transfected into C57BL/6 ES cell line equipped with RMCE docking sites in the ROSA26 locus. Recombinant clones were isolated using neomycin resistance selection and positive clones harbouring six different shRNAs targeting PCF11 were pretested for knockdown potency in ES cells (by qPCR analysis with the following gene expression assay IDs (ThermoFischer); Pcf11: Mm01324032_m1 (exon 1–2) and Mm01324024_m1 (exon 9–10); housekeeper: 4352339E). The clone with highest knockdown efficiency was used for the generation of the mouse line.

All animal experiments were approved by local authorities, and animals' care was in accordance with institutional guidelines.

**In vitro differentiation of murine neurons**. To assay the effect of PCF11 on neuronal differentiation, primary murine neurons were harvested from PCF11 KD (animals) and litter control embryos (E18) by using a neuronal tissue dissociation kit (Miltenyi Biotec). Thereafter, 5000 cells were seeded onto coverslips with primary murine astrocytes as feeder cell (plated 24 h before) in 24-well plates in duplicates. The effect of PCF11 depletion on neurodifferentiation was assessed 4 days after addition of doxycycline (1 µM) to neuronal precursors obtained from PCF11 KD and litter control embryos. To that end, cells were fixed, permeabilised and stained with a primary- (TUBB3) and a Cy3-labelled secondary antibody (as above). Finally, the samples were mount onto microscope slides with Vectashield Antifade Mounting Medium (Vectashield) and the extent of neuronal differentiation was calculated by applying the Neurite Tracer plugin of ImageJ.

**Extraction of TREND signatures from microarray data**. In order to assay the relative proportion of transcript isoforms with shortened or elongated 3′ ends in the GEO GSE49710 data set[44] we selected probes of the relevant microarray platform that could distinguish different TREND isoforms of the same gene. To do so sequences of all probes of Agilent-020382 Human Custom Microarray 44k were mapped to the human genome (version GRCh38/hg38) using STAR[78]. Resulting mappings were overlapped exclusively with tandem sites obtained from the TREND annotation to associate probes with TREND isoforms detected in our experimental setup. Raw microarray data were downloaded from GEO (GSE49710[44]), background-corrected and quantile-normalised using bioconductor limma package[79]. Only genes with probes that could detect at least two different TREND isoforms were used for further analysis. Lengthening index was calculated by dividing expression levels of any detectable long isoform by the levels of the shortest isoform. Lengthening index was used for Student's t-test (Fig. 6b, c), ROC curves and area under curve (AUC) calculations (Fig. 7a, b). AUCs were compared using statistical tests based on bootstrapping and DeLong's method[80].

**Cox modelling**. Cox proportion hazards model was built with 0.7 of training data using MYCN or ALK expression as independent variables. Alternatively, ratios of proximal-to-distal (APA) isoforms of neurodifferentiation operon were used in multivariate Cox model. Modelling and validation was bootstrapped 100 times, concordance (C)-index for model validations were plotted. To assay statistical difference between models, a two-sided t-test was used.

**Methods proteomics**. SDS-PAGE and protein digestion: Forty micrograms of protein was mixed with NuPAGE LDS buffer (Novex) and loaded onto a 4–12% NuPage gel (Invitrogen). Gels were run at 180 V and stained with Instant Blue Coomassie (expedeon). Each lane was cut into 10 slices per lane, which were de-stained, alkylated with 2-iodoacetamide and digested with trypsin, as previously described[81]. Peptides were extracted from the gel pieces with acetonitrile, loaded onto STAGE tips for storage and eluted from the tips shortly before mass spectrometry (MS) analysis[81].

Mass spectrometry: By using an EASY-nLC 1000 (Thermo Scientific) LC system, peptides were separated at a flow rate of 400 nl/min on a self-packed column (75 µm ID, 1.9 µm Reprosil-Pur 120 C-18AQ beads, Dr Maisch Germany) housed in a custom-built column oven at 45 °C. Peptides were separated using gradient of buffers A (0.1% formic acid) and B (80% acetonitrile and 0.1% formic acid): 0–10 min 10% B, 10–55 min 10–38% B, 55–60 min 38–60% B, 60–65 min 60–95% B, 65–70 min 95% B, 70–73 min 95–3% B and 73–75 min 3% B. The column was interfaced with a Nanospray Flex Ion Source (Thermo Scientific) to a Q-Exactive HF mass spectrometer (Thermo Scientific). MS instrument settings were: 1.5 kV spray voltage, full MS at 60 K resolution, AGC target 3e6, range of 300–1750 m/z, max injection time 20 ms, Top 15 MS/MS at 15 K resolution, AGC target 1e5, max injection time 25 ms, isolation width 2.2 m/z, charge exclusion +1 and unassigned, peptide match preferred, exclude isotope on, dynamic exclusion for 20 s.

Protein identification and analysis: Mass spectra were recorded with Xcalibur software 3.1.66.10 (Thermo Scientific). Proteins were identified with Andromeda by searching against human proteome database (71,985 proteins including isoforms) downloaded from UniProt and were quantified with the LFQ algorithm embedded in MaxQuant version 1.5.3.17[63]. The following parameters were used: main search maximum peptide mass error of 4.5 ppm, tryptic peptides of minimum six amino acids length with maximum two missed cleavages, variable oxidation of methionine, protein N-terminal acetylation, fixed cysteine carbamidomethylation, LFQ minimum ratio count of 2, matching between runs enabled, PSM and (Razor) protein false discovery rate of 0.01, advanced ratio estimation and second peptides enabled. Protein-protein interaction network analysis of validated TREND-affected candidates was carried out with String-DB (http://string-db.org/).

## Data availability

Processed TRENDseq data are available at the TREND-DB web explorer [http://shiny.imbei.uni-mainz.de:3838/trend-db]. Raw sequencing and processed TRENDseq data (underlying Figs. 1c, 2b, d, and Supplementary Figs. 2a-d, Supplementary Figs. 3a-d and Supplementary Fig. 5a) is accessible on GEO repository (GSE95057). The source data underlying Fig. 3b and supplementary Fig. 6a can be made available upon reasonable request, and the source data underlying Figs. 5f, g, 6a–c and 7a, b are available in GSE49711 and GSE49710. Source data underlying Supplementary Fig. 7a are available in GSE25219 and data underlying Supplementary Fig. 7d, e are available in GSE49711. Source data underlying Fig. 7c is available in GEO GSE35493, GSE17679 and GSE17679. All other source data can be made available upon reasonable request.

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

## Acknowledgements

The authors would like to express their gratitude to current and former members of the Danckwardt lab; K. Friedemann and A. Canisius for technical assistance; K. Schütze for help with cell cycle analysis; E. Wahle and J. Nourse for critical comments on the manuscript, and P. Cramer, R. Ketting, J. Manley, A. Furger, M. Dutertre, M. Hentze, P. Ivanov, S. Vagner, K. Neugebauer, I. Hollerer, S. Hüttelmaier, S. Köhn, S. Pfister, P. Sorensen, M. Kool and F. Berthold for helpful discussions; and O. Witt, H. Deubzer and the Kulozik lab for providing reagents; C. Meesters from the High Performance Computing Competence Center at the University Mainz and A. Murali for Confocal Microscopy. We apologise to all colleagues whose work could not be discussed or cited here because of space constraints. Work in the laboratory of S.D. is supported by the DFG (DA 1189/2-1), the GRK 1591, by the Federal Ministry of Education and Research (BMBF 01EO1003), by the Hella Bühler Award for Cancer Research and by the German Society of Clinical and Laboratory Medicine (DGKL). M.L. is supported by an Alexander von Humboldt-Bayer Research Fellowship.

## Author contributions

A.O. and S.D. designed research. A.O., S. Ta., S. To., M.H., A.P., M.S. and S.D. performed wet lab experiments, A.O. and M.L. conducted the TRENDseq design and performed bioinformatical analyses. D.S., F.M. and H.B. assembled the TREND-DB. B.T. analysed the neuroblastoma differentiation data. S.M.-G., H.C.P., K.J.L. and F.W. analysed data and discussed results. A.O., M.L. and S.D. analysed the data, discussed the results and wrote the manuscript with critical input from all other contributing authors. S.D. supervised the project.

## Additional information

**Competing interests:** The authors declare no competing interests.

