## [Peer Review File · Nature Communications]

Reviewer #1 (Remarks to the Author):

The manuscript entitled "Transcriptome 3' end organization by PCF11 connects alternative polyadenylation to formation and spontaneous regression of neuroblastoma" by Ogorodnikov and colleagues reports on transcriptome 3' end alterations in the childhood cancer neuroblastoma. A targeted RNAi screen revealed PCF11 as major regulator of 3' end diversification and several relevant pathways seem (de)regulated by this regulator, some of which are associated with clinical outcome of children with neuroblastoma. While various experiments are elegantly designed (e.g. co-depletion of GNB1), several conclusions need to be toned down as mainly speculative in nature.

major comments:

1. The claimed associations with clinical outcome are nothing more than associations; as such, the title should be rephrased and some statements in the abstracts toned down (e.g. "immediate clinical implications"). In line with this, some of the results does no support the conclusions, but are rather speculative (e.g. p15 "... supports an oncogenic function of deregulated APA...").
2. In line with 1, any physiological process may be linked to shorter or longer 3' ends. The authors failed to document that the altered genes are causative. Furthermore, the specificity of their findings to neuroblastoma is not demonstrated (see further).
3. Recent transcriptome studies have indicated pervasive transcription of the human genome, with thousands of long non-coding RNAs expressed. Did the authors explore this class of genes as well and find evidence of alternative polyadenylation?
4. The authors have only studied 3' end diversification and role of PCF11 in neuroblastoma cells. As such it is impossible to assess if the reported findings are neuroblastoma specific or not. Even in neuroblastoma, it is not clear if the phenomenon is confined to MYCN amplified cells or not (as only 2 cell lines from this genetic subgroup are tested in vitro).
5. The authors compared PCF11 levels in metastatic stage 4 neuroblastoma with often regressing 'special' metastatic stage 4S cases. However, as they have not compared with localized stages (1 and/or 2), it cannot be concluded that PCF11 may be linked to spontaneous regression (as PCF11 may also be low in these localized low-risk tumors). In line with this, the authors should also assess transcript lengthening in localized tumors (p14), not only in 4S and 4.
6. PCF11-mediated TREND-signatures are claimed to be 'surprisingly' better for discrimination/classification. This statement should be supported by proper statistics.
7. p19 end 1st paragraph: speculative, needs data
8. The claim that TREND signatures represent 'powerful' biomarkers for neuroblastoma risk stratification is not substantiated. At the very least, proper statistics (e.g. Cox modeling) should be used, with a split training and test cohort, and confirmation in independent cohort.

minor comments:

1. BARD1 is mentioned as one of the few factors affected by somatic alterations in neuroblastoma. MYCN, ALK, PHOX2B, and others are (much) more frequently altered and should also be mentioned.
2. SK-N-BE-2C cells represent only one of the 3 genetic subgroups of neuroblastoma; the authors should clarify this. They also used CHP-134 to some extent, but these cells belong to the same group of high-risk MYCN amplified neuroblastoma.
3. What is 'new' about the 3' end RNA sequencing approach (as the authors state)? Several methods are published and/or commercially available.
4. What is the definition of an ancient gene? Can the authors provide p-value of enrichment of these ancient genes? Same holds true for the enrichment of cancer-associated genes (p-value and definition).
5. Are the 330 proteins (p10) enriched in certain themes or pathways? Is the set of 54 proteins enriched in other themes than 'neurodifferentiation'?
6. What type of murine neurons are used (p13)? Notably, neuroblastoma cells derive from sympathetic nervous system precursor cells.
7. p15: "dominant nature of PCF11", compared to what?

8. p16: is the short or long GNB1 transcript associated with outcome?
9. p17: what is the applied classification definition of low/high risk (neuroblastoma)? Where are subgroups/stages of neuroblastoma included in this analysis?
10. Discussion states that TREND regulators result in 'global' transcriptome 3' end organization. However, authors earlier report a few 100's regulated genes per regulator.
11. Brain is used in discussion as a reference, but neuroblastoma (sympathetic nervous system) has nothing to do with brain (central nervous system) (p19 top).
12. There is no information on the normalization of the RNA seq data.
13. TaqMan is a brand name. Authors should adhere to MIQE guidelines for reporting qPCR experiments.

Reviewer #2 (Remarks to the Author):

The manuscript by Ogorodnikov et al., beautifully tracks the mechanism (function-to-phenotype) of the 3'-end processing machinery in cultured cells to prognosis of outcome in neuroblastoma. In a tour de force, the 3'-end dynamics associated with neuronal differentiation are linked to the level of PCF11 expression. The resultant lengthening of key transcripts that impinge on the WNT signaling pathway cause cell cycle exit and promote neuronal differentiation upon depletion of PCF11. These effects are paralleled in culture, in xenograft models, and in human disease. Remarkably, the differentiation switch is attributable to a single 3'UTR (lengthening of GND1). Loss of just this 3'UTR-isoform promotes de-differentiation in vitro and is associated with poor prognosis in tumours.

To my mind, this study provides the most comprehensive and best-supported evidence to-date linking altered 3'-end processing to cellular transformation and holds direct relevance to human disease progression.

In addition to the comprehensive research study documented in the manuscript, the authors provide a valuable resource to the community via their Shiny app for interactive data interrogation.

Bravo!

Minor comments to improve the manuscript.

TEXT

1. Overall the language was sometimes stilted, a native English-speaking editor could help with that.
2. The sentence "PCF11 acts as a dominant repressor of APA at distal sites (i.e. lengthening of transcriptome)", is confusing, perhaps "PCF11 promotes proximal site choice" is more accurate. Loss of PCF11 promotes lengthening of the transcriptome so call it a repressor seems counter-intuitive.
3. The language around 3'-end lengthening could be made more consistent throughout.
4. The sentence "they also illustrate that such functionally most relevant alterations at the transcriptome 3' end are prone to escape conventional gene expression profiling" on page 20 makes no sense. Remove "such"
5. The word "exon" in "next generation RNA sequencing has led to the discovery of a perplexing complex metazoan transcriptome architecture arising from the alternative use of transcription start sites, exon and introns and polyadenylation sites" is missing an "s".

FIGURES

- Fig 1 a - the image says $\geq 70\%$ but the text says up to 70%. I.e, incompatible concepts.
- Fig 1 b - Indicate what is stained in the images (also other IF images).
- Fig 3 a - Does the size of the dot indicated anything in particular?
- Fig 4 e - consider removing the linear correlation lines, the data don't fit very well. PCF11 KD

especially looks as though the proliferation might be dynamically changed over time (days). Is this due to cell-death?

Fig 6 e - It would be lovely to see the inverse experiment, using PAS-site skipping antisense oligonucleotides to occlude the proximal site, forcing the cells to make the long GNB1 isoform. This would be predicted to drive differentiation (?).

Fig 7 d - I found the schematic confusing. Shouldn't the line from PCF11 go to the left-hand-side of Pol II so that the proximal site is used in high PCF11 and the distal in low PCF11? Perhaps the problem is that it looks as though the polymerase is being inhibited and the aspect of transcriptional-rate and cleavage choice has not been assessed here.

Replies to the reviewer:

We feel honored that both reviewers found merits in the submitted work and observed that our manuscript in a **“tour de force” “beautifully tracks the mechanism (function-to-phenotype) of the 3’-end processing machinery in cultured cells to prognosis of outcome in neuroblastoma”** (reviewer #2), that **“various experiments are elegantly designed (e.g. co-depletion of GNB1)”** (reviewer #1) and that the **“study provides the most comprehensive and best-supported evidence to-date linking altered 3’-end processing to cellular transformation and holds direct relevance to human disease progression”** (reviewer #2). In addition to the **“comprehensive research study”** it was found that the manuscript **“provide(s) a valuable resource to the community via their Shiny app for interactive data”** (reviewer #2).

The main criticism was voiced by reviewer 1, who opines that we overstated the clinical implications of our work. Indeed, our work has been primarily of experimental nature. Thus, while we feel that our findings point to interesting clinical implications that warrant further exploration, we agree that we possibly overstated the clinical perspective in our manuscript. We sincerely believe that clinical scientists have the right expertise to carry further the medical implications much better than we could do. We also believe that the observations made here could serve as a precipitation point for numerous other clinical studies, which are not restricted to neuroblastomas. As suggested by Reviewer 1 we have carefully reworded our manuscript to avoid overstating our clinical observations but presenting them as a starting point for further clinical studies. We are convinced that our revised manuscript strikes a much better balance in presenting our experimental work to the multi-disciplinary readership of *Nature Communications* while still conveying its clinical implications to medical scientists as a basis for further clinical studies.

Reviewer #1 (Remarks to the Author):

The manuscript entitled “Transcriptome 3’end organization by PCF11 connects alternative polyadenylation to formation and spontaneous regression of neuroblastoma” by Ogorodnikov and colleagues reports on transcriptome 3’ end alterations in the childhood cancer neuroblastoma. A targeted RNAi screen revealed PCF11 as major regulator of 3’ end diversification and several relevant pathways seem (de)regulated by this regulator, some of which are associated with clinical outcome of children with neuroblastoma. While various experiments are elegantly designed (e.g. co-depletion of GNB1), several conclusions need to be toned down as mainly speculative in nature.

Response:

While we wanted to highlight the clinical implications that may arise from our study, we agree that our study is largely experimental work and more clinical studies are required to fully assess the clinical

implications. As suggested we have rewritten these paragraphs, toning down these conclusions (detailed below).

major comments:

1. The claimed associations with clinical outcome are nothing more than associations; as such, the title should be rephrased and some statements in the abstracts toned down (e.g. “immediate clinical implications”). In line with this, some of the results does no support the conclusions, but are rather speculative (e.g. p15 “... supports an oncogenic function of deregulated APA...”).

Response:

We agree with these very important remarks and have changed the wording of the title and last sentence of the abstract and page 15 as follows:

Title: “Transcriptome 3’end organization by PCF11 links alternative polyadenylation to formation and neuronal differentiation of neuroblastoma”

Abstract, (previously) last sentence: “Our findings document a critical role for APA in tumourigenesis and describe a novel mechanism for cell fate reprogramming in neuroblastoma with potentially important clinical implications.”

Introduction, last paragraph: “... we discover an unexpected critical role for PCF11-dependent APA regulation in neuronal differentiation with potentially important implications for spontaneous tumour regression.

Page 15: “... suggests an oncogenic function of deregulated APA...”

2. In line with 1, any physiological process may be linked to shorter or longer 3’ ends. The authors failed to document that the altered genes are causative. Furthermore, the specificity of their findings to neuroblastoma is not demonstrated (see further).

Response:

We are glad that we can directly clarify this important point of critique - as it concerns one of the most central aspects of the entire paper, including the causality and all downstream considerations whether or not our finding of deregulated TREND is likely more than a mere association with tumor risk and possibly spontaneous tumor regression. As such it also directly affects several points below mentioned by this reviewer.

We respectfully disagree with the notion that “the authors failed to document that the altered genes are causative”. We clearly show that a PCF11-depletion phenotype (resulting in longer transcripts isoforms) leads to neurodifferentiation (in murine and human samples) and that this effect can be fully antagonized by changing the relative proportion of alternative transcript isoforms of one single gene, namely GNB1 (Fig. 6e and f). This directly demonstrates that the relative abundance of long and short GNB1 transcript isoforms are causative in regulating neurodifferentiation.

This reviewer explicitly mentions this experiment (co-depletion of GNB1) to be “elegantly designed” in his/her introductory statement (which is also in line with the statement of reviewer #2: “ ... remarkably, the differentiation switch is attributable to a single 3’UTR...”). It is of utmost importance to realize that this experiment (Fig. 6 e, f together with Fig. 6d) was exactly carried out in order to further explore whether the ‘mere’ association of GNB1 lengthening with high and low risk and survival observed in patients before (Fig. 6c) is likely causal and functionally relevant.

We have analyzed some other tumor entities and could not find a GNB1 causality, however we believe that a more comprehensive study would be necessary. It needs to be noted, however, that this process does not have to be specific for neuroblastoma to become relevant (see loss of function of TP53 and other oncogenic events across various tumor entities).

*In order to make the causality issue (see first part of the comment) more explicit we have changed the wording on page 17 accordingly: “This directly corroborates the functional importance of PCF11-directed APA-regulation, in which alterations of GNB1 transcript isoforms appear to play a central **and causative** role”.*

3. Recent transcriptome studies have indicated pervasive transcription of the human genome, with thousands of long non-coding RNAs expressed. Did the authors explore this class of genes as well and find evidence of alternative polyadenylation?

Response:

This is an interesting question. We are currently looking into other RNA species (including lncRNAs). Preliminary data suggest some regulation, yet in light of the in parts complicate (and different) biogenesis of lncRNAs these studies require additional and other types of controls, before any solid statement can be made. Thus any statement concerning lncRNAs, at present, would be merely speculative. In addition our data clearly show causality for the GNB1 transcript isoforms (see reply to comment #2 above and Fig. 6e, f), which is mirrored by the findings in neuroblastoma patients (Fig. 6b, c). If at all, modification of lncRNAs may thus be an additional layer of regulation. We decided not to consider this for discussion as it appeared somewhat speculative and beyond the scope of the present manuscript.

4. The authors have only studied 3’ end diversification and role of PCF11 in neuroblastoma cells. As such it is impossible to assess if the reported findings are neuroblastoma specific or not. Even in

neuroblastoma, it is not clear if the phenomenon is confined to MYCN amplified cells or not (as only 2 cell lines from this genetic subgroup are tested in vitro).

Response:

To prove the generality of PCF11 for neuronal differentiation (beyond that observed in neuroblastoma BE(2)-2 and CHP-134 cells) we generated (1) a new transgenic mouse model, which allows to specifically deplete PCF11. Based on neuronal precursors obtained from this animal we also observed and report a lengthening phenotype for murine central precursor neurons upon down-modulation of PCF11 (Fig. 5 d, e). (2) Further we also noted that high and low PCF11 levels are associated with overall shorter and longer transcript isoforms with matured and immature brain human brain samples (supplements). (3) In line with this, our own ongoing studies show that PCF11 depletion leads to lengthening of transcriptome in various other tissues (not shown). This suggests that PCF11 is very likely not a neuroblastoma specific regulator of TREND and that this regulation is not confined to MYCN amplified cells.

Consistent with these observations, PCF11 regulated TREND is confined to tissues expressing PCF11 (not shown). Among those neuronal cells appear to express somewhat higher levels, with particularly high levels in immature (malignant and non-malignant) neuronal cells (Fig. 7c, Supplementary Figure 7a, b).

This has now been made more explicit in the revised manuscript (page 13, first paragraph) by implementing a new passage:

“Thus, although neuroblastomas derive from sympathetic nervous system precursor cells, it appears that they share neurodevelopmental features with neurons in the central nervous system with PCF11-dependent APA regulation being an important mechanism in this process”.

and by modifying the concluding sentence of the following paragraph (page 13, last paragraph):

“Thus our data indicate that this phenomenon is not confined to MYCN amplified cells nor restricted to neuroblastoma. This suggests a more global role of PCF11 in coordinating the timely switch to fully committed neuronal fate (Fig. 4g and Fig. 5e)... “

5. The authors compared PCF11 levels in metastatic stage 4 neuroblastoma with often regressing ‘special’ metastatic stage 4S cases. However, as they have not compared with localized stages (1 and/or 2), it cannot be concluded that PCF11 may be linked to spontaneous regression (as PCF11 may also be low in these localized low-risk tumors). In line with this, the authors should also assess transcript lengthening in localized tumors (p14), not only in 4S and 4.

Response:

We thank the reviewer for this very important statement and would also like to refer to our reply to comment #1 (and our assessment of where this experimental paper is primarily positioned). We agree

with the reviewer that exploring the underlying mechanism of spontaneous regression is by far more complex. This is also reflected by the overall very sparse number of (clinical) papers shedding light onto this enigmatic tumor phenotype (see below).

Throughout the paper we show that PCF11-downregulation leads to TREND lengthening and that low PCF11 levels are associated with lengthening of a neurodifferentiation operon. We further show that downregulation of PCF11 induces neurodifferentiation (in human cells and murine neuronal precursors). Further we identify that high and low PCF11 levels are associated with metastatic stage 4 neuroblastoma and often regressing 4S cases. Looking into the identify of transcripts differentially affected by TREND we even retrieve 17 out of 26 detectable TREND-affected genes belonging to the neurodifferentiation module identified in the PCF11-depletion setup (Fig. 3b, 4a), including AES, IGF1R and GNB1 (Fig. 6b).

Although there are conceptionally not too many approaches for analyzing the underlying mechanisms of spontaneous regression in a clinical setting (Brodeur & Bagatell Nat Rev Clin Oncol 2014), the comparison between stage 4 and stage 4S neuroblastoma is not uncommon and has been used in several setups to obtain insights into this enigmatic 4S phenotype (for example:

- Benard J, et al. MYCN-non-amplified metastatic neuroblastoma with good prognosis and spontaneous regression: a molecular portrait of stage 4S. *Mol Oncol.* 2008;2:261–271.
- Lavarino C, et al. Specific gene expression profiles and chromosomal abnormalities are associated with infant disseminated neuroblastoma. *BMC Cancer.* 2009;9:44.
- Yu F, et al. Proteomics-based identification of spontaneous regression-associated proteins in neuroblastoma. *J Pediatr Surg.* 2011;46:1948–1955).

However, owing to the inherent difficulties of this type of analysis and the important remark of this reviewer we now chose to describe our findings more carefully as an ‘association’:

“...Thus low-level PCF11 is **associated** with a better outcome and **possibly** a greater likelihood for spontaneous tumour regression. This reflects our *in vitro* and *in vivo* observations (Fig. 4c-g, 5a-e and **Supplementary Figure 6a-e**) and is corroborated by the respective expression signature of established markers for spontaneous neuroblastoma regression (that is a higher expression of HOXC948 and CHD549, Fig. 5g)....” and concluded that “**further studies are required to more comprehensively illuminate the role of PCF11 for spontaneous regression**”.

Owing to this comment and comment # 1 also brought up by this reviewer we also decided to replace “spontaneous regression” by “neuronal differentiation” in the title and tone down the respective passages

in the introduction and discussion section. We believe that this more appropriately reflects the main observations.

We thus fully concur with the objection of this reviewer that further analyses are required to more comprehensively clarify the spontaneous regression issue. However we believe that this lies beyond the scope of the present manuscript.

6. PCF11-mediated TREND-signatures are claimed to be ‘surprisingly’ better for discrimination/classification. This statement should be supported by proper statistics.

Response:

We thank the reviewer for pointing out this omission and regret this mistake. Our statement is now supported by statistics showing significant P-values of bootstrap comparison between the predictive power of established risk marker expression and combined TREND-patterns (illustrated in the table below and implemented into the paper as Supplementary Table 7)

P-values of bootstrap comparison between the predictive power of established risk marker expression and combined TREND-patterns

Stratifier	Risk marker	All samples	Non MYCN amp samples
Death	MYCN	$1.6 * 10^{-4}$	$3.1 * 10^{-8}$
Death	ALK	$7.9 * 10^{-5}$	$1.2 * 10^{-3}$
High Risk	MYCN	$2.6 * 10^{-7}$	$4.0 * 10^{-14}$
High Risk	ALK	$2.3 * 10^{-9}$	$2.3 * 10^{-5}$

7. p19 end 1st paragraph: speculative, needs data

Response:

Concerning this comment we also want to refer to our comment to point #4 and the respective passages in the paper where we directly show that PCF11 downregulation induces neuronal differentiation in a mouse model (Fig. 5 d, e), and that high and low PCF11 levels are associated with overall shorter and longer transcript isoforms with matured and immature brain human (and murine) brain samples (supplements).

We agree that this point in the discussion section is merely speculative. However we felt that this may have important implications and thus discussed our findings in an integrative manner. Owing to the critique we reworded this piece of the discussion section as follows:

“... Neuronal PCF11 expression drops around birth and during neuronal differentiation, but appears to be high in neuroblastomas and, interestingly, other paediatric cancer entities with embryonic origin (**Fig. 7c**). Thus, **it is tempting to speculate that** sustained (postnatal) PCF11 expression may drive highly proliferative embryonic programs by arresting cells in an undifferentiated state. This could also explain the high frequency of microscopic neuroblastic lesions in fetuses or young infants compared to older ages ...”. and now added that **“...although future studies are needed to dissect the role of PCF11 for embryonic development in further detail...”**.

8. The claim that TREND signatures represent ‘powerful’ biomarkers for neuroblastoma risk stratification is not substantiated. At the very least, proper statistics (e.g. Cox modeling) should be used, with a split training and test cohort, and confirmation in independent cohort.

Response:

We fully agree (see also comment to point #6, above), the reviewer raises an important point and we apologize that we failed to support this by statistics.

We now included statistics supporting that PCF11 might represent a powerful biomarker (see comment to point #6). We implemented an additional supplementary table showing significant P-values of bootstrap comparison between the predictive power of established risk marker expression and combined TREND-patterns. We fully concur with this reviewer that more in-depth analysis are required in order to test the suitability of PCF11 as biomarker. However we believe that reliably testing this goes beyond the scope of this paper and would primarily require independent cohorts. In light of the overall very limited number of patients and hence limited access to patient material and/or studies, we further believe that these studies have to be carried out in larger neuroblastoma research consortia, which are sufficiently powered for in depth statistical analysis. Ultimately it is important to realize that even already available RNAseq data for existing patient cohorts does not qualify for most of the TREND analysis (owing to the limited resolution of differential 3’end read count mappings).

*Accordingly (see also comment to point #6), we added **“that there are further studies needed in order to validate TREND as diagnostic biomarkers in larger and independent patient cohorts”**.*

minor comments:

1. BARD1 is mentioned as one of the few factors affected by somatic alterations in neuroblastoma. MYCN, ALK, PHOX2B, and others are (much) more frequently altered and should also be mentioned.

Response:

We apologize that we did not mention this in the original version of the manuscript (as we felt that this disrupts the flow of this section). This important bit of information has now been included in the introduction.

2. SK-N-BE-2C cells represent only one of the 3 genetic subgroups of neuroblastoma; the authors should clarify this. They also used CHP-134 to some extent, but these cells belong to the same group of high-risk MYCN amplified neuroblastoma.

Response:

We thank the reviewer for pointing out this aspect; this has now been made explicit at page 13/14:

“...PCF11-depletion in primary murine neurons obtained from the central nervous system of these animals led to neurodifferentiation (Fig. 5e), which is consistent with the neurodifferentiation phenotype upon depletion of PCF11 in the BE(2)-C and CHP-134 model system (Fig. 4g, Supplementary Figure 6c). Thus, although BE(2)-C and CHP-134 represent only one of the three genetic subgroups of neuroblastoma (i.e. high risk MYCN amplified tumors), our data indicate that this phenomenon is not confined to MYCN amplified cells nor restricted to neuroblastoma. This suggests a more global role of PCF11 in coordinating the timely switch to fully committed neuronal fate (Fig. 4g and Fig. 5e) thereby preventing uncontrolled embryonic proliferative programs that may eventually give rise to neuroblastic tumours”.

3. What is ‘new’ about the 3’ end RNA sequencing approach (as the authors state)? Several methods are published and/or commercially available.

Response:

We developed TRENDseq for highly multiplexed TREND sequencing, which was required for the large screening study covering >160 TREND regulators (Fig. 2). Although this screening has been started more than 6 years ago, this protocol - to the best of our knowledge - is still one of the few (if not the only one) that permits multiplexed 3’end seq with reasonable resolution.

We added a clarification to make this more explicit (page 7).

4. What is the definition of an ancient gene? Can the authors provide p-value of enrichment of these ancient genes? Same holds true for the enrichment of cancer-associated genes (p-value and definition).

Response:

Ancient genes are genes whose gene age is dated to more than 450 Million years (referring to Supplementary Figure 2b,c). The color-key for the p-values is indicated in the figure (last row of heatmap; Supplementary Figure 2c). Only p-values below 0.05 are colored in the heatmap.

The list of cancer-associated genes (referring to inlet shown in Fig. 2c) is obtained from COSMIC (cancer.sanger.ac.uk, PMID: 27899578). All p-values are shown in the already existing Supplementary Table 3 (the enrichment p-value for PCF11 in cancer is 2,98E-09). This has now been made more explicit in the text (page 8) and in the legend of the respective figures (Fig. 2c and Supplementary Figure 2b,c).

5. Are the 330 proteins (p10) enriched in certain themes or pathways? Is the set of 54 proteins enriched in other themes than 'neurodifferentiation'?

Response:

This is shown in the paper (Supplementary Figure 6a and Fig. 3). Beyond themes such as neurodifferentiation, axonogenesis, axon guidance, neuron projection morphogenesis and development, cell morphology there is RNA metabolism and various signaling pathways (including WNT but also pathways such as G-beta-gamma signaling, EIF2 signaling, amyloid processing, unfolded protein response).

6. What type of murine neurons are used (p13)? Notably, neuroblastoma cells derive from sympathetic nervous system precursor cells.

Response:

These are primary murine neurons obtained from the central nervous system of these animals. This has now been made clear in the text:

"... PCF11-depletion in primary murine neurons obtained from the central nervous system of these animals led to neurodifferentiation (Fig. 5e), which is consistent with the neurodifferentiation phenotype upon depletion of PCF11 in the BE(2)-C and CHP-134 model system (Fig. 4g, Supplementary Figure 6c).

Owing to this very important statement we also implemented a passage (at the end of the preceding paragraph) that makes more explicit the origin of these different cell types:

"Thus although neuroblastomas derive from sympathetic nervous system precursor cells, it appears that they share neurodevelopmental features with neurons in the central nervous system with PCF11-dependent APA regulation being an important mechanism in this process." (page 13).

7. p15: "dominant nature of PCF11", compared to what?

Response:

Compared to other potential APA regulators that could also play an important role in neurodifferentiation of Neuroblastomas (Fig.2, Fig.3). We have now clarified this in the revised version (page 15/16, last paragraph)

8. p16: is the short or long GNB1 transcript associated with outcome?

Response:

We thank the reviewer for asking this critical question as this passage could indeed be misunderstood. The relative lengthening is associated with better outcome. This has been made more explicit (page 15, and page 17).

As a remark on that: the absolute abundance of either the one or the other isoform alone does not have strong predictive potential which suggests that there is a functional interaction between both transcript isoforms (by so far not identified mechanisms in trans, be it by RNA-binding proteins or other components). Coming from the RNA biology perspective this is a very exciting finding as it potentially implies novel 3'UTR-mediated processes (templated by Hüttelmaier S et al Nature 2005 or Berkovits & Mayr C. Nature 2015). However this currently under intense investigation and any statement on that, at the moment, would be merely speculative.

9. p17: what is the applied classification definition of low/high risk (neuroblastoma)? Where are subgroups/stages of neuroblastoma included in this analysis?

Response:

The classifications stem from the patient data sheet (GSE49710). Basically all stages of all 498 patients are included in the analysis, however only the binary classifier is visualized in the ROC curve (death/alive or high/low risk).

10. Discussion states that TREND regulators result in 'global' transcriptome 3' end organization. However, authors earlier report a few 100's regulated genes per regulator.

Response:

We thank this reviewer for pointing our attention to this aspect. In total, we detected > 3 600 differentially TREND-affected genes to be significantly regulated (p. 7). Considering the total number of TREND-isoforms detected (20 156) we retrieved approximately half of them (9 168) being regulated upon one or the other knockdown (Fig. 2b, top panel).

The average number of TREND-affected genes was 130 per regulator. Importantly, however, the vast majority of 'regulators' depleted in this study are related to mechanisms that do not have any direct (and/or so far established direct/indirect) function in the formation of mRNA 3'ends. Thus it is not entirely surprising (and even expected) that there are dozens of regulators with relatively mild effects. In contrast,

when considering those regulators that are directly connected to RNA metabolism or, even more, specifically to 3' end processing we identify several hundred genes (up to 1400) regulated per regulator. However, because of the multiplexing in TRENDseq we can only obtain information for the most abundant transcripts, making these numbers rather conservative estimates (taking into account that >75% genes are affected by APA, and given a retrieval rate of 50% of them being dynamically regulated through APA (see above) the affects reported here may even represent a drastic underestimation of the actual affect). We now made more explicit that there are several regulators controlling substantial numbers (hundreds) of genes, which justifies to talk about 'global' transcriptome 3' end organization much better than in the original version. We modified this passage in the results and discussion section accordingly.

11. Brain is used in discussion as a reference, but neuroblastoma (sympathetic nervous system) has nothing to do with brain (central nervous system) (p19 top).

Response:

In response to this point we also would like to refer to the comment to point #7. Although neuroblastomas clearly belong to the sympathetic nervous system (and not to the central nervous system), we find it quite remarkable that there are apparently some interesting parallels, which in terms of neurodifferentiation are even experimentally substantiated in the present manuscript:

(1) We see a differentiation of neuronal precursors upon depletion of PCF11 (Fig. 5e), (2) we observe the same for neuroblastomas in vitro (Fig. 3a, 6f), and see (3) a similar association for stage 4 and stage 4S neuroblastomas in patients (Fig. 5g), respectively. At the same time we found (4) that neuronal PCF11 expression drops around birth in humans (suppl. Fig. 7a) and during differentiation in mice (suppl. Fig. 7b, c). (5) Further high level of PCF11 expression can be found in other pediatric cancer entities with embryonic origin.

This led us to sum up and discuss these observations as follows (page 20, 1st paragraph):

"...Neuronal PCF11 expression drops around birth and during neuronal differentiation, but appears to be high in neuroblastomas and, interestingly, other paediatric cancer entities with embryonic origin (**Fig. 7c**). Thus, sustained (postnatal) PCF11 expression may drive highly proliferative embryonic programs by arresting cells in an undifferentiated state. This could also explain the high frequency of microscopic neuroblastic lesions in fetuses or young infants compared to older ages ...".

and added "...although future studies are needed to further dissect the role of PCF11 for embryonic development in detail..." to avoid that this can be misunderstood (as we believe further studies are required to fully address this topic).

In addition (see also our own comment above) we added a sentence in the result section that makes this difference more explicit and better explains what we can deduce from this comparison:

“Thus although neuroblastomas derive from sympathetic nervous system precursor cells, it appears that they share neurodevelopmental features with neurons in the central nervous system with PCF11-dependent APA regulation being an important mechanism in this process.” (page 13).

12. There is no information on the normalization of the RNA seq data.

Response:

This is an important question. As TREND- and APA-events are internally controlled (this is the number of reads at an annotated pA site per gene in relation to reads at a second pA site in exactly the same gene; see also tandem pA array assays, for example: Danckwardt et al Blood 2004; Danckwardt et al EMBO J 2006; Gehring et al Nat Genet. 2001) a normalization in the classical sense is initially not required (unlike conventional RNA seq or array data analysis). Accordingly we essentially report relative changes of the expression level of each transcript isoform per gene. The longest significantly affected (BH-adjusted $p \leq 0.05$) and annotated transcript isoform expressed by the respective gene served as normalizer (both in terms of directionality i.e. shortening or lengthening and abundance change) to compare 2 (or more) experimental conditions.

In the method section (page 24/25) we describe this in more details as follows:

“...Mapped reads were filtered from internal priming events using the assembly of TREND annotation (see above). Number of reads associated with each TREND-isoform was calculated using HTSeq (htseq-count command, intersection-strict option). Number of reads aligned to each site represent expression of individual 3' end isoform.

For statistical analysis, the expression level of each transcript isoform was examined by Fisher's exact test in comparison to the respective other alternative 3'end isoforms expressed by the same gene. Contingency table included the number of reads of tested isoform and total amount of reads of all the other isoforms of the gene (for the knockdown and control samples, respectively). Obtained p-values were adjusted using Benjamini-Hochberg method, and adjusted p value ≤ 0.05 filter was applied. To calculate fold-regulation per isoform, total amount of reads for each gene was normalized to 100%, and percentage of individual isoform in the knockdown sample was divided by the percentage of the same isoform in the control.

*The most 3'end position and fold-change of regulated transcript isoforms (**Fig. 2b** and **Supplementary Figure 3c**) was calculated relative to the “Zero-isoform”, which is defined as the longest significantly affected (BH-adjusted $p \leq 0.05$) and annotated transcript isoform expressed by the respective gene. To describe the overall tendency of a given gene to express shortened (or lengthened, respectively) transcript isoforms, a proxy of two most significantly affected isoforms was applied (**Supplementary Figure 2c, Fig. 3b, Supplementary 3b, d**). The shortening index was calculated as the fold-regulation of the shorter*

isoform normalized to the fold-regulation of the longer isoform of the same gene (a positive log2 shortening index represents a higher abundance of the shorter transcript isoform upon depletion of the respective TREND-regulator, and vice versa)...”.

13. TaqMan is a brand name. Authors should adhere to MIQE guidelines for reporting qPCR experiments.

Response:

We thank the reviewer for this comment and changed this passage accordingly.

In conclusion, we very much appreciate the critical comments of this reviewer as it helped streamlining the clinical outlook of present manuscript. We strongly believe that this paper and the inferred implications represent a very important starting point for further exploring independent clinical cohorts including neuroblastomas and possibly other tumor entities.

Reviewer #2 (Remarks to the Author):

The manuscript by Ogorodnikov et al., beautifully tracks the mechanism (function-to-phenotype) of the 3'-end processing machinery in cultured cells to prognosis of outcome in neuroblastoma. In a tour de force, the 3'-end dynamics associated with neuronal differentiation are linked to the level of PCF11 expression. The resultant lengthening of key transcripts that impinge on the WNT signaling pathway cause cell cycle exit and promote neuronal differentiation upon depletion of PCF11. These effects are paralleled in culture, in xenograft models, and in human disease. Remarkably, the differentiation switch is attributable to a single 3'UTR (lengthening of GND1). Loss of just this 3'UTR-isoform promotes de-differentiation in vitro and is associated with poor prognosis in tumours.

To my mind, this study provides the most comprehensive and best-supported evidence to-date linking altered 3'-end processing to cellular transformation and holds direct relevance to human disease progression.

In addition to the comprehensive research study documented in the manuscript, the authors provide a valuable resource to the community via their Shiny app for interactive data interrogation.

Bravo!

Response:

We thank the reviewer for this very enthusiastic statement. Going through our manuscript and the comment above, we felt it might make sense to more explicitly refer to our interactive data resource.

Accordingly we now modified the abstract by adding a final sentence stating:

“We provide an interactive data repository of transcriptome-wide APA covering >170 RNAs, and an APA-network map with regulatory hubs”.

Minor comments to improve the manuscript.

1. Overall the language was sometimes stilted, a native English-speaking editor could help with that.

Response:

We thank the reviewer for bringing this to our attention. We went through the manuscript and reworded the text to improve clarity.

2. The sentence "PCF11 acts as a dominant repressor of APA at distal sites (i.e. lengthening of transcriptome)", is confusing, perhaps "PCF11 promotes proximal site choice" is more accurate. Loss of PCF11 promotes lengthening of the transcriptome so call it a repressor seems counter-intuitive.

Response:

We agree this is indeed somewhat confusing. However it is important to realize that the statement in this particular context refers to the dual-depletion experiments (aiming to disentangle the functional hierarchy of several APA-regulators). In fact we believe that PCF11 carries out its function on poly(A)site choice primarily via its role in fostering transcription termination (and not through its somewhat disputed role as a 'bona fide' 3'end processing factor). As such it represses processing that could otherwise occur further downstream. We also have indications that the PCF11 action in terms of proximal processing events might be even dispensable and that a PCF11-dependent effect can be mimicked pharmaceutically.

We took this comment, however, to clarify the role of PCF11 in plain words in a concluding sentence of a preceding paragraph (to make this more accessible for a broad readership):

"Thus in (undifferentiated) neuroblastoma PCF11 promotes proximal polyadenylation site choice, while CPSF6 and NUDT21 facilitate processing at distal polyadenylation sites".

We also modified the following paragraph referred to above (highlighted):

*"PCF11 acts as a dominant repressor of APA at distal sites (**i.e. promotes processing at proximal sites**), and counteracts the repression of APA at proximal sites executed by CPSF6.*

3. The language around 3'-end lengthening could be made more consistent throughout.

Response:

We thank the reviewer for pointing this out. We changed the wording to improve clarity.

4. The sentence "they also illustrate that such functionally most relevant alterations at the transcriptome 3' end are prone to escape conventional gene expression profiling" on page 20 makes no sense. Remove "such"

Response:

We agree this does not make sense. This has been corrected accordingly.

5. The word "exon" in "next generation RNA sequencing has led to the discovery of a perplexing complex metazoan transcriptome architecture arising from the alternative use of transcription start sites, exon and introns and polyadenylation sites" is missing an "s".

Response:

We corrected this mistake

FIGURES

Fig 1 a - the image says $\geq 70\%$ but the text says up to 70%. I.e, incompatible concepts.

Response:

We thank the reviewer for pointing out this obvious discrepancy. This has been modified accordingly.

Fig 1 b – Indicate what is stained in the images (also other IF images).

Response:

We apologize for omitting this. This has now been added in the legend to Fig. 1b (and all subsequent instances).

Fig 3 a – Does the size of the dot indicated anything in particular?

Response:

The slightly larger diameter of the dot denoted 'PCF11' was intended to highlight the lead candidate.

Fig 4 e – consider removing the linear correlation lines, the data don't fit very well. PCF11 KD especially looks as though the proliferation might be dynamically changed over time (days). Is this due to cell-death?

Response:

The regression was implemented to highlight the overall tendency. This is getting probably clearer in the coloured figures.

Importantly, IPTG (to induce the shRNA-mediated knockdown of PCF11) was added on day 0, and hence the delay until full knockdown is the most likely explanation for this retarded yet very drastic effect from day 5 on. Unfortunately an experimental setup in the reverse order (first induction of knockdown, followed by seeding of and counting cells) is not applicable as PCF11 deficiency led to an adherence defect, making a proliferation study in this order impossible.

Although it is difficult to directly compare the kinetics of our proliferation assay (Fig. 4e) with those in the apoptosis assay (Fig. 4f) and the effect on cell cycle progression (Fig. 4d), we would consider cell-death to be indeed a very important component (apart from differentiation of cells that do not longer proliferate).

Fig 6 e - It would be lovely to see the inverse experiment, using PAS-site skipping antisense oligonucleotides to occlude the proximal site, forcing the cells to make the long GNB1 isoform. This would be predicted to drive differentiation (?).

Response:

This is a very important comment. In fact we also carried out overexpression studies (of long and short isoforms respectively). Although this does not exactly resemble the same type of experiment proposed (a forced expression of long isoforms also gives rise to shorter isoforms) we observed some differentiation, which however was by far not complete. On the basis of these (and other preliminary) experiments we are currently speculating that GNB1-lengthening is a necessary but not sufficient component of PCF11 mediated neurodifferentiation. This would also be consistent with our observation that we find an entire neurodifferentiation operon (Fig. 3b) including 54 genes, and not just GNB1 regulated upon PCF11 depletion. However, this clearly needs to be further elucidated in the future.

Fig 7 d – I found the schematic confusing. Shouldn't the line from PCF11 go to the left-hand-side of Pol II so that the proximal site is used in high PCF11 and the distal in low PCF11? Perhaps the problem is that it looks as though the polymerase is being inhibited and the aspect of transcriptional-rate and cleavage choice has not been assessed here.

Response:

We thank the reviewer for pointing this out.

We have now decided to further explain our “integrated model for PCF11-dependent TREND-regulation in neuroblastoma governing neurodifferentiation” in the legend. We felt it would be a lot more convenient – as it also helps the reader to understand the scheme independently (without going back to the text). Although this is not (yet) self-explanatory (due to the complexity of the topic) we believe this makes the model a lot more accessible. It also refers to the relevant literature that originally describes the effect of PCF11 on transcription termination.

REVIEWERS' COMMENTS:

Reviewer #1 (Remarks to the Author):

I'm satisfied with the revised version of the manuscript. The authors did a good job in improving the manuscript in terms of clarity and aligning conclusions with results.

Reviewer #2 (Remarks to the Author):

The authors have systematically addressed my comments and concerns. I was already convinced of the manuscript's importance on first reading and remain an advocate. I appreciate the care that has been taken with this revision and look forward to seeing it published. Congratulations again on a fine body of work!